# High-throughput 5′ UTR engineering for enhanced protein production in non-viral gene therapies

Jicong Cao [1,2,3,4,9], Eva Maria Novoa[4,5,6,7,9], Zhizhuo Zhang[4,5,6,9], William C. W. Chen[1,2,3], Dianbo Liu[1,4,5], Gigi C. G. Choi[1,2,3,8], Alan S. L. Wong[1,2,3,8], Claudia Wehrspaun[1,2,3], Manolis Kellis [4,5,6✉] & Timothy K. Lu [1,2,3,4,6✉]

Despite significant clinical progress in cell and gene therapies, maximizing protein expression in order to enhance potency remains a major technical challenge. Here, we develop a high-throughput strategy to design, screen, and optimize 5′ UTRs that enhance protein expression from a strong human cytomegalovirus (CMV) promoter. We first identify naturally occurring 5′ UTRs with high translation efficiencies and use this information with in silico genetic algorithms to generate synthetic 5′ UTRs. A total of ~12,000 5′ UTRs are then screened using a recombinase-mediated integration strategy that greatly enhances the sensitivity of high-throughput screens by eliminating copy number and position effects that limit lentiviral approaches. Using this approach, we identify three synthetic 5′ UTRs that outperform commonly used non-viral gene therapy plasmids in expressing protein payloads. In summary, we demonstrate that high-throughput screening of 5′ UTR libraries with recombinase-mediated integration can identify genetic elements that enhance protein expression, which should have numerous applications for engineered cell and gene therapies.

[1] Synthetic Biology Group, Research Laboratory of Electronics, Massachusetts Institute of Technology, Cambridge, MA, USA. [2] Department of Biological Engineering, Massachusetts Institute of Technology, Cambridge, MA, USA. [3] Synthetic Biology Center, Massachusetts Institute of Technology, Cambridge, MA, USA. [4] Broad Institute of MIT and Harvard, Cambridge, MA, USA. [5] Computer Science and Artificial Intelligence Laboratory, Massachusetts Institute of Technology, Cambridge, MA, USA. [6] Department of Electrical Engineering and Computer Science, Massachusetts Institute of Technology, Cambridge, MA, USA. [7] Present address: Center for Genomic Regulation (CRG), Barcelona, Spain. [8] Present address: School of Biomedical Sciences, University of Hong Kong, Hong Kong, China. [9] The authors contributed equally: Jicong Cao, Eva Maria Novoa, Zhizhuo Zhang. ✉email: manoli@mit.edu; timlu@mit.edu

In recent years, gene therapies that enable the exogenous production of proteins to replace defective genes have started to have a transformative clinical impact[1–3]. Gene therapies can be delivered into patients through viral vectors or non-viral vectors. One of the major challenges facing current gene therapy approaches is maximizing potency, since increasing the amount of exogenously expressed protein can reduce dose requirements and thus manufacturing costs, while improving human clinical results[4]. Multiple strategies are being employed to improve potency, such as enhancing cellular transduction efficiency by using more efficient viral vectors or non-viral transduction reagents, or by improving the gene expression construct itself. For example, recombinant adeno-associated virus (rAAV) is one of the major modalities used in gene therapy due to its infectivity and ability to achieve long-term gene expression in vivo[5,6]. However, $10^{16}$–$10^{17}$ genome copies (GCs) of rAAVs are being used in clinical trials, which requires the use of 100–10,000-L scale bioreactors to produce exceptionally large amounts of cGMP-grade viruses and is expensive[4,7]. Furthermore, a recent non-human primate study showed that the administration of high-dose intravenous rAAVs can lead to severe liver and neuronal toxicity[8]. Thus, there is a significant unmet need to improve protein production from viral gene therapies to enlarge the therapeutic window and reduce costs.

In addition, enhancing gene expression for non-viral DNA therapy remains a significant challenge. Non-viral gene therapy delivers DNA into cells to produce therapeutic proteins or vaccine antigens in vivo, with several potential advantages over viral gene therapies[9–11]. First, non-viral DNA therapy is potentially less immunogenic than viral particles since it uses chemical delivery strategies[12–14]. Second, the ability to deliver large amounts of DNA cargo via non-viral routes is greater than with viral vectors, where packaging limits place significant restrictions. Third, plasmids are relatively inexpensive to produce at the research and industrial scale and are more stable than viruses[15,16]. The efficiency of in vivo DNA delivery has improved significantly as a result of recent advancements in liposome chemistry and nanoparticles, but is still not as efficient as viruses in many cases[17–19]. Thus, repeat dosing or increased dose levels have been attempted, but these strategies can incur greater costs and risk of side effects and can sacrifice patient convenience.

In addition to optimizing delivery efficiencies, protein expression from gene therapies can be enhanced by optimizing the nucleic acid payload being delivered. Gene expression cassettes consist of multiple elements: promoter (which may include an enhancer), 5′ untranslated region (5′ UTR), protein-coding region, 3′ UTR, and polyadenylation (PolyA) signal[20]. Previous work has involved promoter engineering to enhance transcription or to enable cell-type specific gene expression[21,22]. However, fewer efforts to modulate translation through UTR engineering have been described.

Here, we focus on optimizing the 5′ UTR to improve protein production in a non-viral gene therapy context. The rational design of 5′ UTRs to enhance protein expression remains challenging, even though regulatory elements and 5′ UTR sequences that regulate gene expression in certain scenarios have been identified[23–27]. The design of 5′ UTRs has been held back by limited knowledge of the relationships between 5′ UTR sequences and associated levels of protein expression. For example, Sample et al.[27] screened a library of 50-bp random 5′ UTRs including the Kozak sequence in the randomized region, and found that 5′ UTRs that include strong Kozak sequences lead to higher ribosome recruitment. However, plasmids commonly used for non-viral gene therapy, such as pVAX1, already contain strong Kozak sequences. Thus, we were not interested in exploring the region that comprised the Kozak sequence as a diversity region but,

rather, whether variability in the 5′ UTR region preceding a strong Kozak sequence could lead to the identification of 5′ UTRs that would enhance protein expression levels.

In this study, we develop a platform to systematically screen and engineer 5′ UTRs that can enhance protein expression in mammalian cells (Fig. 1). We first identify naturally occurring 5′ UTRs with different translational activities in multiple human cell types. We then apply a genetic algorithm to obtain synthetic 5′ UTRs, which were generated by evolving strong endogenous human 5′ UTRs in silico. To enable high-throughput testing of 12,000 distinct 5′ UTRs, we develop a recombinase-based library screening strategy to eliminate copy number artefacts and positional effects, which introduce significant noise in traditional lentiviral-based library screening approaches[28,29]. Through this approach, we identify three synthetic 5′ UTRs that significantly outperformed both naturally occurring 5′ UTRs, as well as a plasmid (pVAX1) that is commonly used for non-viral gene therapy[30]. Finally, we show that the three synthetic 5′ UTRs enhance protein expression across a variety of cell types and that

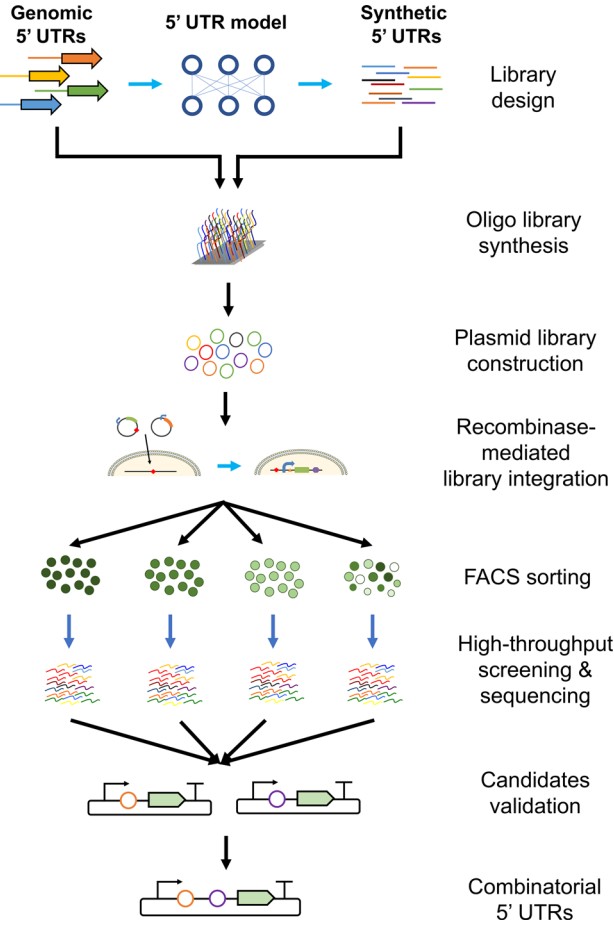

**Fig. 1 Schematic overview of recombinase-mediated 5′ UTR library screening strategy.** Naturally occurring 5′ UTRs were extracted, analyzed, and used as the training set to generate synthetic 5′ UTRs for screening. Oligos encoding the 5′ UTR library were synthesized and cloned into plasmids containing a recombinase-recognition site and a GFP reporter. The resulting plasmids were transfected into the HEK 293T-LP cell line with the corresponding recombinase recognition site, resulting in targeted genomic insertion. The cells were sorted into bins based on GFP intensities, and the 5′ UTR sequences of each bin were amplified, sequenced, counted, and compared. The 5′ UTR candidates that enhanced GFP expression were selected and validated experimentally. Finally, the top-ranked validated 5′ UTRs were combined to test for increased gene expression.

these synthetic 5′ UTRs can be combined to improve protein expression levels, thus highlighting the potential of this approach for gene therapy applications.

## Results

**5′ UTR model training and library design by genetic algorithms.** Protein production comprises two major steps: in the first step, DNA is transcribed into mRNA; and in the second step, mRNA is translated into protein. While transcription and translation are coupled in prokaryotic cells, these two steps are uncoupled in eukaryotic cells. Consequently, eukaryotic protein expression levels are highly dependent on mRNA levels, which are governed by the transcription machinery, but also on the translation efficiency (TE) of the transcripts, which is governed by the translation machinery[31,32]. In this context, given an identical transcription rate for two transcripts, the differences in the final amount of protein can be modulated by features found in the 5′ UTR regions, which are involved in the recruitment of ribosomes[33]. The TE of a gene, i.e., the rate of mRNA translation into protein, can be calculated as the ratio of the ribosomal footprints (RPF) observed on a given mRNA of interest, which can be measured using Ribo-seq[34], to the relative abundance of that mRNA transcript in the cell, which can be measured using RNA-seq.

We first investigated the TEs of naturally occurring 5′ UTRs (Fig. 2). To this end, we gathered publicly available Ribo-seq and RNA-seq data from human muscle tissue[35], as well as from two human cell lines, human embryonic kidney (HEK) 293T[36] and human prostate cancer cell line PC3[37]. A 5′ UTR length of 100 bp was chosen and fixed for training algorithms and engineering 5′ UTRs, which is compatible with the limits of current commercially available ssDNA template biosynthesis. For 5′ UTR sequences that were longer than 100 bp, sequences were extracted from the 5′ end and 3′ end to construct two 100-bp long 5′ UTRs; those shorter than 100 bp were filled up with repeats of a CAA motif that does not have known secondary structure[38] to create two sequence versions, one having a shift of one nucleotide relative to the other (see "Methods"). AUGs were removed by randomly mutating one of the three nucleotides to avoid generating undesired upstream open reading frames (Supplementary Data 1).

Next, we computationally generated synthetic sequences by mutating and evolving endogenous 5′ UTRs in silico. We trained and developed a computational model to predict TE based on 5′ UTR characteristics (Fig. 2). Specifically, we extracted sequence features of 5′ UTR regions that could be associated with gene expression levels and TE, which included k-mer frequency, RNA folding energy, 5′ UTR length, and number of ORFs (Supplementary Fig. 1). A random forest regression model was then trained on sequence features to predict TE and mRNA expression (Supplementary Fig. 2). The model was trained on experimentally determined TE rates and mRNA levels, which were obtained from analyzing publicly available RNA-seq and Ribo-seq data of endogenous genes from the three human cell types noted above: HEK 293T cells, PC3 cells, and human muscle tissue[39]. We also found that random forest regression showed the highest prediction among four different models, including random forest, glmnet, Rpart, and SVM (Supplementary Figs. 3–5). Given that searching for all $4^{100}$ possible 100-bp sequences would be too computationally demanding, we applied a genetic algorithm[40], which simulates the evolution process, to search for "optimal" sequences by mutating and recombining the endogenous sequences (see Methods). We created 2388 synthetic 5′ UTRs

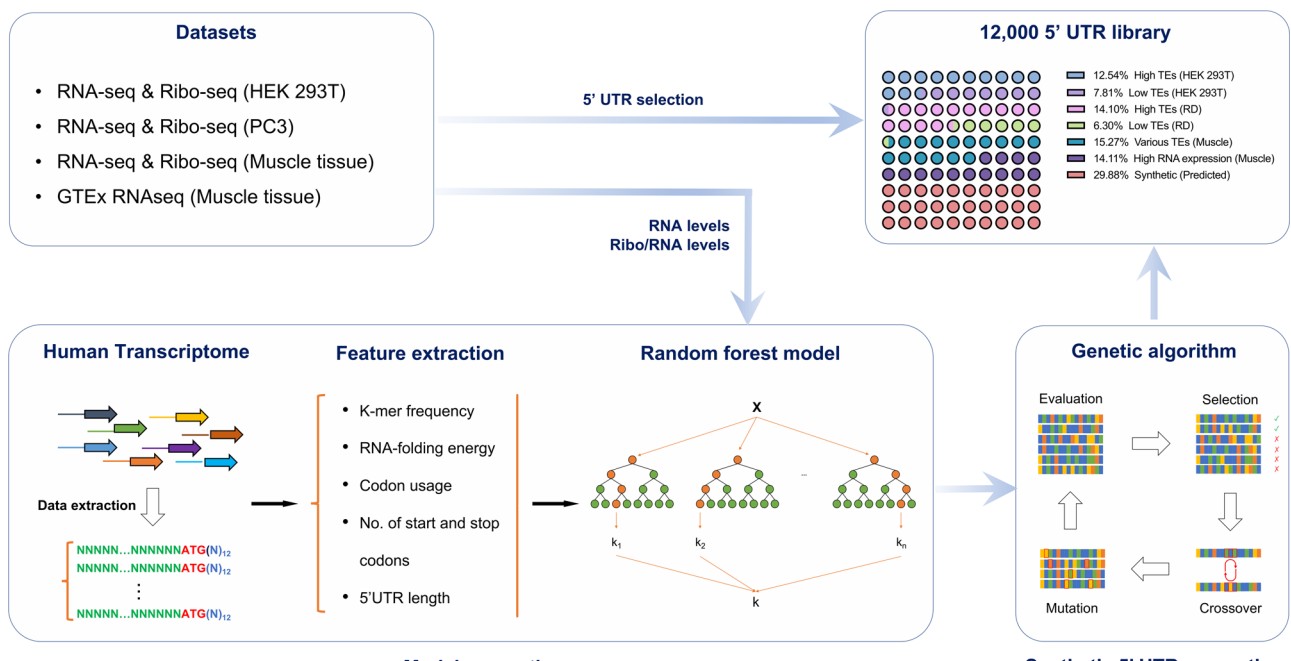

**Fig. 2 Design of the 5′ UTR library of naturally occurring and synthetic 5′ UTRs.** RNA-seq and Ribo-seq datasets of HEK 293T, PC3, and human muscle cells, together with the GTEx database of human muscle tissue, were collected. Natural 5′ UTRs with high TEs and low TEs in HEK 293T and RD cells, 5′ UTRs with various TEs in human muscle cells, and the 5′ UTRs with high mRNA counts in human muscle tissues were selected and added to the library. In addition, we designed synthetic 5′ UTRs by: (i) collecting endogenous 5′ UTR sequences on the target cell type (HEK 293T, PC3 or human muscle cells) from public data; (ii) extracting sequence features of the 5′ UTRs, including those nucleotides surrounding the AUG region; (iii) training a Random Forest machine learning method for each cell type/tissue (HEK 293T, PC3 or human muscle cells), to learn a function that maps sequence features to mRNA expression levels and TEs; and (iv) designing a set of 100 bp synthetic sequences that are predicted to maximize TEs and protein expression levels using genetic algorithms.

that were predicted to have high TEs (Supplementary Data 2), in addition to a testing set designed by evolving 1198 5′ UTRs with a range of TEs within 2 evolutionary generations (Supplementary Data 3). Overall, a total of 3586 synthetic sequences and 8414 naturally occurring sequences were used to build the ~12,000 100-bp 5′ UTR library for this study.

**Recombinase-mediated library screening to minimize copy number and position effects.** Lentiviral-based library screening is the most commonly used method for high-throughput genetic screening[41–43]. In this method, diversified genetic elements are cloned into a lentiviral carrier plasmid and transfected into a virus-producing cell line with packaging and envelope plasmids to produce a lentiviral library, which is then used to infect the cells of interest. A multiplicity of infection (MOI) of ~0.1–0.3 is widely used to ensure that most of infected cells receive only one copy of the element of interest. However, even at 0.1 MOI, 10% of the cells receive two or more copies. Moreover, lentiviruses insert randomly into the cellular genome, resulting in significant variations in gene expression[44,45]. As a result, a significant amount of noise due to copy number variations in cells and positional effects can obscure accurate phenotypic assessment of genetic constructs in lentiviral screens.

To address this issue, we designed a recombinase-based gene integration strategy to screen the 5′ UTR library; this strategy ensures single-copy integration within each cell at a defined "landing-pad" location (Fig. 3a). We used the serine recombinase Bxb1 to integrate a plasmid containing the Bxb1 attB site into the Bxb1 attP site on the genome, which results in destruction of the attP site to prevent additional insertions[46–49].

We first constructed HEK 293T cell lines with a landing pad by lentiviral infection. In these cell lines, the landing pad comprised a constitutive promoter, a mutant BxbI attP site with enhanced integration efficiency[50], and a yellow fluorescent protein (YFP)[47] as a reporter for the integration of the landing pad. Nine cell clones with insertion of the landing pad were identified and expanded. We chose to use two different cell lines with different YFP expression levels during our screens to reduce the impact of genomic location on the screening phenotype. The 5′ UTR library was cloned upstream of the GFP reporter on the payload plasmid, which also encoded a BxB1 attB site and a red fluorescent protein (RFP) and puromycin duo selection marker (Fig. 3b). In this system, successful integration activates the expression of RFP and puromycin and inhibits YFP expression. We integrated the 5′ UTR library into the two different landing pad cell lines with >25-fold more cells than the size of the library (>300k integrated cells). The transfected cells were grown for 3 days, then subjected to puromycin selection for 1 week.

To identify 5′ UTRs with increased protein expression, we used FACS to sort the cell library into four bins based on GFP expression levels: top 2.5%, 2.5–5%, 5–10%, and 0–100% (unsorted). We then extracted genomic DNAs from cells in each bin and optimized PCR conditions for unbiased amplicon amplification[41]. The amplicons were then barcoded and sequenced using Illumina NextSeq. We calculated the relative abundance of each 5′ UTR sequence in each of the three top bins (2.5%, 2.5–5%, and 5–10%) and normalized them to the counts in the control bin (0–100%). Log2 ratios were used to represent the enrichment of each 5′ UTR in each bin.

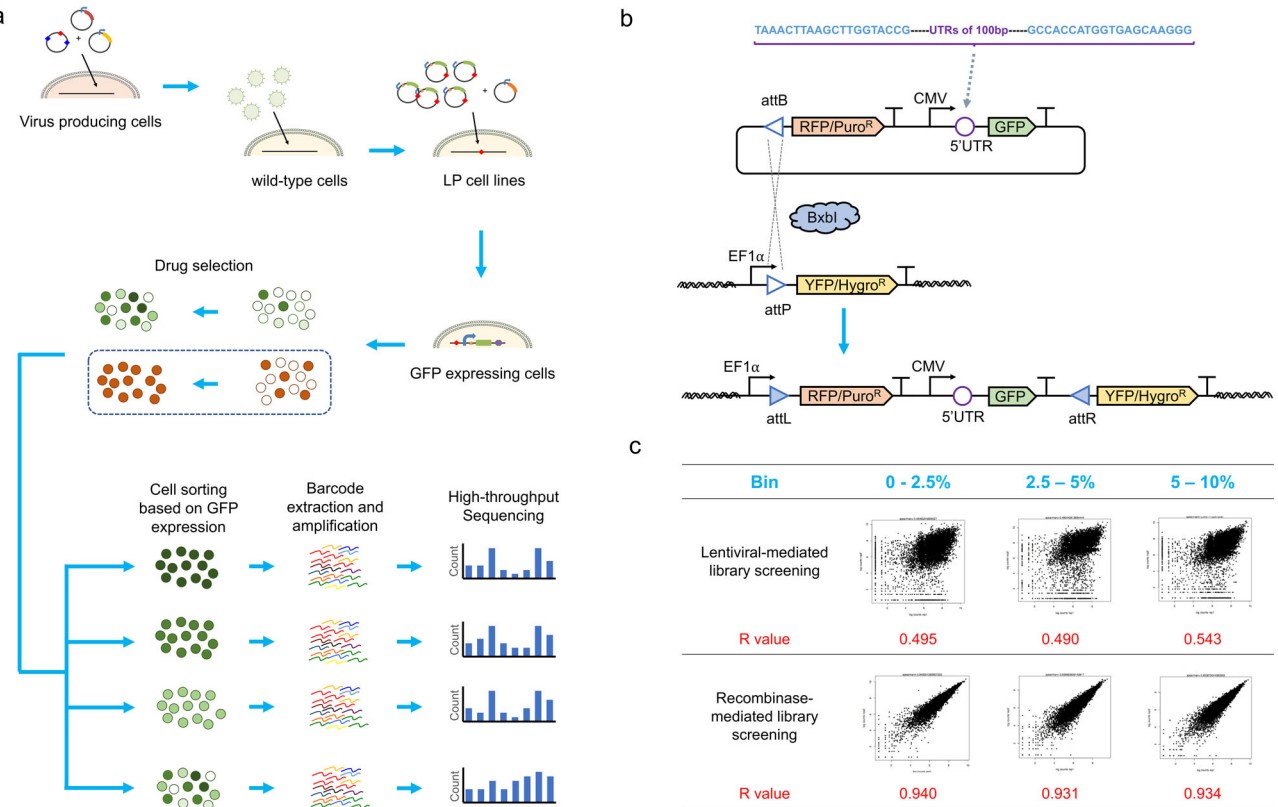

**Fig. 3 Strategy for constructing HEK 293T cell lines with a landing pad and screening the 5′ UTR library using recombinase-based gene integration. a** Recombinase-based library screening workflow. **b** Construction of the 5′ UTR library and schematic illustration of recombinase-based gene integration. **c** We observed high reproducibility for barcode representations between two HEK-LP cell lines independently transfected with the library and a recombinase-expression plasmid; cells were sorted into three bins based on GFP expression (top 0–2.5%, top 2.5–5%, and top 5–10%). log2 values of normalized barcode counts are shown. R is the Pearson correlation coefficient.

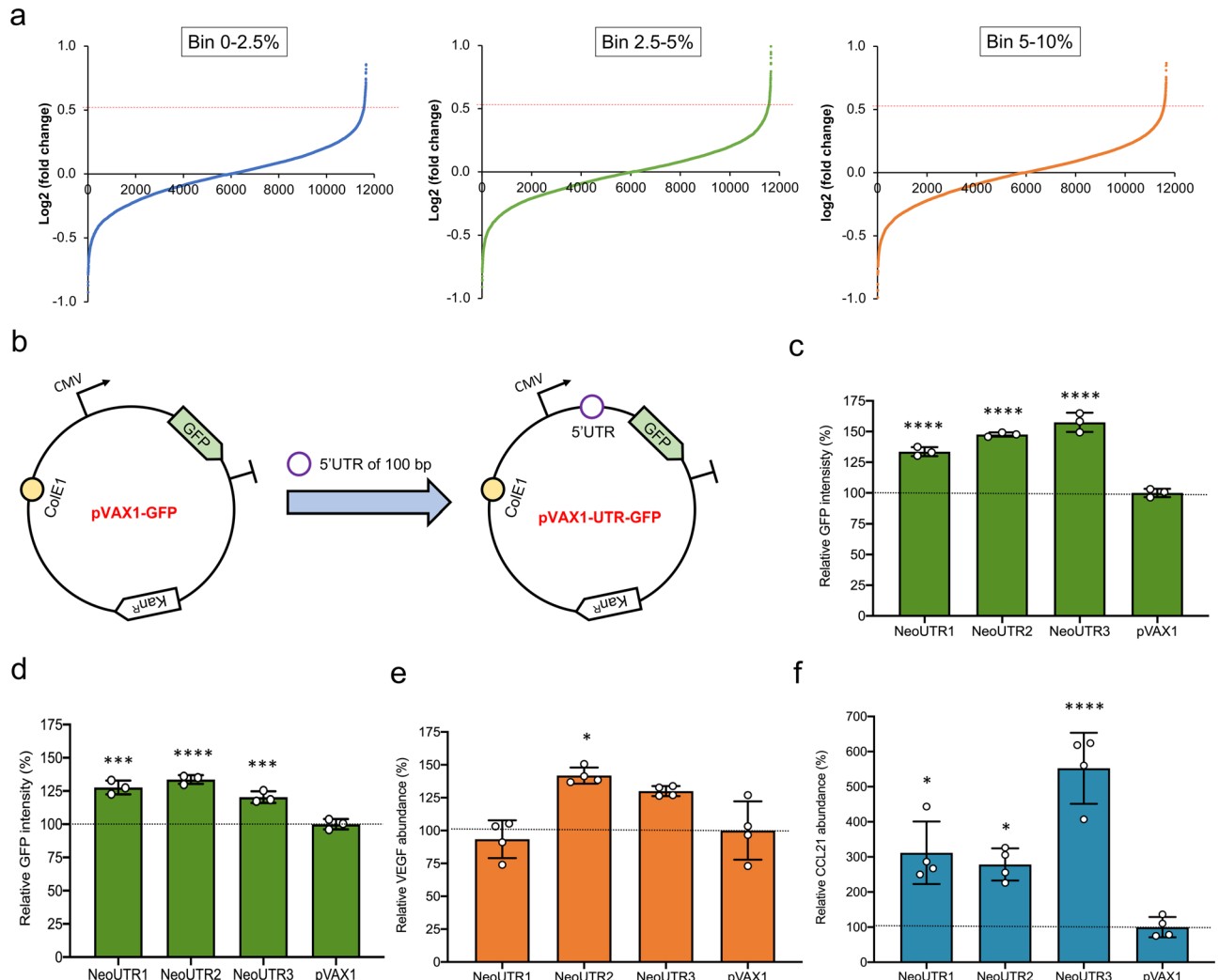

**Fig. 4 Selection and validation of 5′ UTR candidates. a** 5′ UTRs that modulate protein expression were ranked by their mean log2 ratios (compared with the control of unsorted cells) of the normalized barcode count in the three bins based on GFP expression. 5′ UTRs with a log2 ratio greater than 0.52 (which is highlighted as a red dotted line) in all three bins were selected for further validation. **b** The GFP gene was inserted into the pVAX1 plasmid to make the pVAX1-GFP plasmid, which was used as a control in the GFP expression study. 5′ UTR candidates were inserted directly upstream of the Kozak sequence of the GFP coding sequence to make the pVAX1-UTR-GFP plasmids. **c** Three 5′ UTR candidates that significantly enhanced protein expression were chosen for further testing. The *p*-values for NeoUTR1, NeoUTR2, NeoUTR3 vs pVAX1 are <0.0001, <0.0001, and <0.0001. **d** The effects of the three 5′ UTRs on GFP expression in RD cells. The *p*-values for NeoUTR1, NeoUTR2, NeoUTR3 vs pVAX1 are 0.0001, <0.0001 and 0.0010. **e** The effects of the three 5′ UTRs on VEGF expression in RD cells. The *p*-values for NeoUTR1, NeoUTR2, NeoUTR3 vs pVAX1 are 0.8838, 0.0146 and 0.0675. **f** The effects of the three 5′ UTRs on CCL21 expression in RD cells. The *p*-values for NeoUTR1, NeoUTR2, NeoUTR3 vs pVAX1 are 0.0183, 0.0412, and 0.0002. Relative protein expression in each sample was normalized to that of the pVAX1 plasmid (relative expression (%) = 100 is highlighted as a gray dotted line). Source data are provided as a Source data file. Statistical differences between groups were analyzed by ordinary one-way ANOVA with 95% confidence interval. Data are presented as mean values ± SD for three biological replicates (**c**, **d**) or four biological replicates (**e**, **f**). (*$p < 0.05$; **$p < 0.01$; ***$p < 0.001$; ****$p < 0.0001$ vs pVAX1).

Our results showed that this recombinase-based library screening approach achieved Pearson correlation values greater than 0.93 between results obtained from the two landing pad cell lines in all three bins, thus demonstrating the high reproducibility of the screening process (Fig. 3c). This level of reproducibility exceeded that of traditional lentiviral-based library screening, which had correlation values equal to 0.49, 0.49, and 0.54 for each of the three bins, respectively. Overall, our results clearly show that recombinase-based integration can significantly improve the reproducibility of high-throughput screening.

**Validation of the 5′ UTR hits in HEK 293T cells**. To select candidates for further experimental validation, we ranked the

5′ UTRs based on their relative expression levels in top expressing bins (2.5%, 2.5–5%, and 5–10%), relative to the unsorted bin. Specifically, differentially enriched 5′ UTRs in the top-expressing bins were determined using DESeq2[51], which takes into account variability across biological replicates to identify differentially expressed candidates. Top candidates were defined as 5′ UTRs that showed at least 50% increased expression level in all three top-expressing bins (fold change greater than 50%) with significant adjusted *p*-values in all three bins (*p*-adj < 0.05) (Supplementary Data 4–6), relative to control (Fig. 4a). Using these criteria, thirteen 5′ UTRs with enriched expression in all three bins were used for further validation. Interestingly, six of these thirteen 5′ UTRs were synthetic 5′ UTRs, implying a 300% enrichment towards synthetic sequences predicted to have

high TEs (six out of 2388) compared to the pool of natural 5′ UTRs included in the screening (seven out of 8414).

We then tested the selected 5′ UTR candidates in the pVAX1 non-viral gene therapy plasmid[30]. pVAX1 has a human CMV promoter for high-level protein expression, a multiple cloning site for foreign gene insertion, and a bovine growth hormone (bGH) PolyA signal for transcriptional termination. We synthesized and inserted the thirteen candidate 5′ UTRs (100 bp long) along with a green fluorescent protein (GFP) reporter (with Kozak sequence; pVAX1-UTR-GFP) downstream of the CMV promoter in the pVAX1 plasmid. As a control, we used only the GFP reporter (with Kozak sequence; pVAX1-GFP) (Fig. 4b). We co-transfected HEK 293T cells with the engineered 5′ UTR-containing plasmids and the control plasmid along with a blue fluorescent protein (BFP) expression plasmid. This allowed us to normalize GFP expression to transfection efficiency, as determined by BFP levels. Six out of the thirteen tested plasmids showed higher GFP expression than the commercial protein expression plasmid pVAX1 in HEK 293T cells (Supplementary Fig. 6), and all thirteen plasmids showed higher GFP expression than the median of the library. These results not only validated the effectiveness of the screening but also demonstrated that our approach can successfully identify 5′ UTRs that outperform at least one plasmid that is widely used and optimized for gene therapy.

**Synthetic 5′ UTRs enhance protein expression levels in non-viral DNA therapies.** From the six experimentally validated 5′ UTRs that increased GFP production in HEK 293T cells (Supplementary Fig. 6), we chose the top three 5′ UTRs candidates for additional testing in human muscle cells, since DNA-encoded therapeutics are often delivered into human muscle to trigger the expression of vaccines or therapeutics[52,53] (Fig. 4c), Notably, all three 5′ UTRs (NeoUTR1, NeoUTR2, and NeoUTR3), which increased protein abundance of GFP by 37% to 58% in HEK 293T, were designed by our genetic algorithms. We used human rhabdomyosarcoma (RD) cells as a model for human muscle cells, finding that all three 5′ UTRs enhanced GFP expression in RD cells by 34% (Fig. 4d). We also compared the synthetic 5′ UTRs with commonly used introns and with 5′ UTRs that have been used to enhance gene expression under the CMV promoter in previous studies: (i) a chimeric intron from the pmax Cloning plasmid (https://bioscience.lonza.com/lonza_bs/US/en/Transfection/p/000000000000191671/pmaxCloning-Vector), (ii) intron 2 of the human beta-globin gene[54], (iii) a tripartite leader sequence of human adenovirus mRNA linked with a major late promoter enhancer (TM)[55], and (iv) the first intron of the human CMV immediate early gene (Intron A)[56]. Compared to these sequences, our synthetic 5′ UTRs more strongly enhanced gene expression in HEK 293T and RD cells (Supplementary Fig. 7).

To demonstrate the potential therapeutic utility of these UTRs, we expressed two different therapeutic proteins with these 5′ UTRs: vascular endothelial growth factor (VEGF), which stimulates the formation of blood vessels[57]; and C–C motif chemokine ligand 21 (CCL21), which can recruit immune cells for immunotherapy[58]. Two out of the three 5′ UTRs increased VEGF expression compared to the commercial plasmid, and one of these, NeoUTR2, increased VEGF production by 42% relative to pVAX1 ($p = 0.01$) (Fig. 4e). All three 5′ UTRs increased CCL21 expression by greater than 100% relative to pVAX1 ($p = 0.02$, 0.04, 0.0002, respectively), and NeoUTR3 showed an impressive increase of 452% (Fig. 4f). In summary, we identified three 5′ UTR thru in silico design and high-throughput screening that significantly increase protein expression levels for fluorescent protein reporters (by up to 58%) and therapeutic proteins (by up to 452%) from non-viral gene therapy vectors.

**Combinatorial synthetic 5′ UTRs can further enhance protein expression.** Considering our promising results using synthetic 5′ UTRs, we sought to investigate whether combinations of these 5′ UTRs might further enhance GFP expression levels (Fig. 5a). Using our three NeoUTR leads as building blocks, we constructed six combinatorial 5′ UTRs (CoNeoUTRs) by joining two of the NuUTRs with a 6-nt linker (CAACAA). These were labeled as CoNeoUTR2-3 (NeoUTR2-NeoUTR3), CoNuUTR1-3 (NeoUTR1-NeoUTR3), Co NeoUTR3-2 (NeoUTR3-NeoUTR2), CoNeoUTR1-2 (NeoUTR1-NeoUTR2), CoNeoUTR3-1 (NeoUTR3-NeoUTR1), and CoNeo UTR2-1 (NeoUTR2-NeoUTR1). We inserted each combinatorial 5′ UTR upstream of the GFP coding sequence, co-transfected HEK 293T cells with the resulting plasmids and the BFP expression plasmid, and then measured GFP and BFP fluorescence (Fig. 5b). We observed that the strength of the 5′ UTR combinations was positively correlated with the strengths of the two individual 5′ UTRs (Supplementary Fig. 8) Moreover, we observed that for the CoNeoUTRs constructed with two different NeoUTRs, the strength was higher if the stronger NeoUTR was placed at the 3′ end: CoNeoUTR1-2 > CoNeoUTR2-1, CoNeoUTR1-3 > CoNeoUTR3-1, and CoNeo UTR2-3 > CoNeoUTR3-2.

Finally, we tested how the artificial 5′ UTR elements modulate gene expression in different cell types. In addition to HEK 293T, we chose human and mouse muscle cell lines RD and C2C12, respectively, because muscle is a common targeted tissue for vaccines and neutralizing antibody gene therapies[59,60]. We also selected human breast cancer cell line MCF-7 to test its potential applicability in cancerous cell lines for cancer gene therapies[61]. We found that all three 100-bp artificial 5′ UTRs (NeoUTR1, NeoUTR2, and NeoUTR3) enhanced protein expression in the three cell types and HEK 293T cells; however, the relative strengths of the 5′ UTRs were different in different cell types (Fig. 5c). Overall, 78% of all conditions tested, the synthetic 5′ UTRs were statistically stronger than pVAX1 (the p-values are labeled in Fig. 5c) across the four cell types. Thus, these results show that the 5′ UTR sequences and their combinatorial counterparts identified in this study can significantly enhance protein expression across a variety of mammalian cell types, further validating the applicability of our approach.

In summary, synthetic 5′ UTRs can enhance protein production across multiple cell types and can be combined together to further modulate protein levels.

**Discussion**

In this study, we developed a robust strategy for the systematic discovery and engineering of 5′ UTRs for enhanced protein expression. We trained a computational model using gene expression information on naturally occurring 5′ UTRs and evolved a synthetic 5′ UTR library. We developed a recombinase-based high-throughput screening platform to overcome the significant heterogeneity that limits the accuracy of lentiviral-based screens. The serine recombinase BxbI integrates one copy of tagged genetic elements at a specific location in the host genome, eliminating the copy number and position effects that are seen in conventional lentiviral-based library screening. We observed high reproducibility of this recombinase-based library screening strategy for 5′ UTR engineering, allowing us to identify three synthetic 5′ UTR candidates that increase protein production across multiple cell types. This strategy allowed us to identify synthetic 5′ UTRs that outperformed the commonly used pVAX1 vector and four commonly used introns in terms of their ability to increase protein production as non-viral gene delivery vectors.

Previous work has demonstrated that machine learning methods can be employed to predict 5′ UTR translation efficiency in mammalian cells and yeast[24–27]. Here, we extended the use of

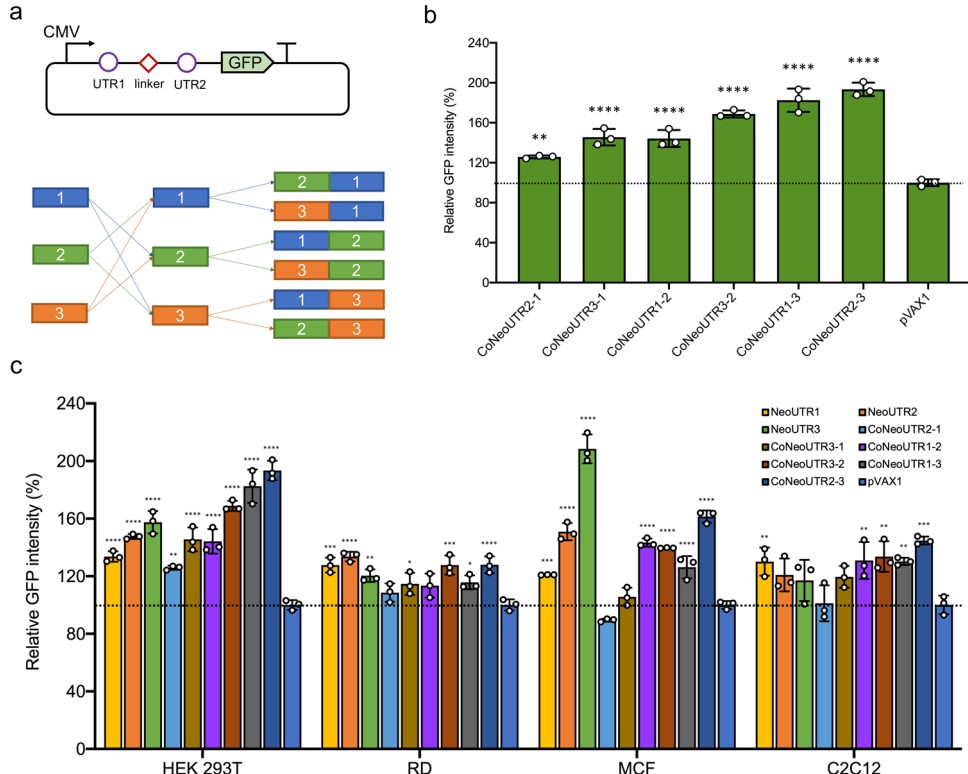

**Fig. 5 Effects of combinatorial 5′ UTRs on GFP expression in various cell lines. a** We constructed six distinct 5′ UTR combinations by combining different pairwise permutations of the three validated 5′ UTR candidates with a CAACAA linker between them, and then inserted these combinations into the pVAX1-GFP plasmid directly upstream of the Kozak sequence. **b** GFP expression from the 5′ UTR combinations on GFP expression in HEK 293T cells. Statistical differences between groups were analyzed by ordinary one-way ANOVA with 95% confidence interval. The relative protein expression was normalized to that from the pVAX1-GFP plasmid, set as 100 (%) and highlighted as a gray dotted line. Data are presented as mean values ± SD for three biological replicates. The *p*-values for CoNeoUTR2-1, CoNeoUTR3-1, CoNeoUTR1-2, CoNeoUTR3-2, CoNeoUTR1-3, and CoNeoUTR2-3 vs pVAX1 are 0.0025, <0.0001, <0.0001, <0.0001, <0.0001, and <0.0001. **c** Test of the single and combinatorial 5′ UTRs on GFP expression in various cell lines. Source data are provided as a Source data file. Statistical differences between groups were analyzed by two-way ANOVA with 95% confidence interval. Dunnett test was performed to correct for multiple comparisons in ANOVA post-hoc analysis. The relative protein expression was normalized to that from the pVAX1-GFP plasmid, set as 100 (%) and highlighted as a gray dotted line. Data are presented as mean values ± SD for three biological replicates. In HEK 293T cell lines, the *p*-values for NeoUTR1, NeoUTR2, NeoUTR3, CoNeoUTR2-1, CoNeoUTR3-1, CoNeoUTR1-2, CoNeoUTR3-2, CoNeoUTR1-3, and CoNeoUTR2-3 vs pVAX1 are <0.0001, <0.0001, <0.0001, 0.0002, <0.0001, <0.0001, <0.0001, <0.0001, and <0.0001. In RD cell lines, the p-values for NeoUTR1, NeoUTR2, NeoUTR3, CoNeoUTR2-1, CoNeoUTR3-1, CoNeoUTR1-2, CoNeoUTR3-2, CoNeoUTR1-3, and CoNeoUTR2-3 vs pVAX1 are <0.0001, <0.0001, 0.0055, 0.5989, 0.0836, 0.1336, <0.0001, 0.0549, and <0.0001. In MCF cell lines, the p-values for NeoUTR1, NeoUTR2, NeoUTR3, CoNeoUTR2-1, CoNeoUTR3-1, CoNeoUTR1-2, CoNeoUTR3-2, CoNeoUTR1-3, and CoNeoUTR2-3 vs pVAX1 are 0.0042, <0.0001, <0.0001, 0.3513, 0.9132, <0.0001, <0.0001, 0.0002, and <0.0001. In C2C12 cell lines, the *p*-values for NeoUTR1, NeoUTR2, NeoUTR3, CoNeoUTR2-1, CoNeoUTR3-1, CoNeoUTR1-2, CoNeoUTR3-2, CoNeoUTR1-3, and CoNeoUTR2-3 vs pVAX1 are <0.0001, 0.0042, 0.0306, 0.9997, 0.0089, <0.0001, <0.0001, <0.0001, and <0.0001. (*$p < 0.05$; **$p < 0.01$; ***$p < 0.001$; ****$p < 0.0001$ vs pVAX1).

machine learning from prediction to the de novo design of 5′ UTRs that would potentially enhance protein expression in human cells, which imposes a key challenge in gene therapy and drug manufacturing. Compared with elucidating mechanisms for validated 5′ UTRs, such as the key role of Kozak sequences[27] or harpins near the 5′ cap[62,63], our work instead focus on identifying 5′ UTRs that are stronger than that on the commercially and clinically used gene therapy vectors and the commonly used introns and other 5′ UTRs. Future work could include investigations into the underlying mechanisms that enhance gene expression from these 5′ UTRs to provide insights for further improvements.

Lentiviral-based library screening is often a common choice for massive parallel reporter assays. Here, we observed that lentiviral-based 5′ UTR screenings yielded poor reproducibility across biological replicates (Fig. 3c). To overcome this limitation, we constructed a recombinase-based library screening platform, which we find eliminates the copy number and position effects in

conventional lentiviral-based screening, and is virtually applicable for screening of any genetic element of interest. Our library of 12,000 5′ UTRs was originally synthesized in the form of an array of 100-bp long DNA oligonucleotides and subsequently cloned into vectors (Fig. 1). However, we should note that naturally occurring 5′ UTRs in mammalian species are often longer than 100 bp, Future work exploring 5′ UTR diversity might benefit from improved DNA synthesis technologies that will make it possible to expand the length of the diversity region that is tested in the screening. In this regard, here we found that individual synthetic 5′ UTRs can be combined to enhance protein expression. In the future, these regions could be combined with translation initiation site (TIS) sequences[64] that might potentially outperform the commonly used Kozak sequences for gene therapy.

Although the synthetic 5′ UTRs we validated in this study were strong across multiple contexts, their relative performance did vary depending on cell type. To optimize gene therapy for specific

types of cells, it could be useful to repeat the strategy employed here but focused on the cells of interest. In addition, future work should test whether these synthetic 5′ UTRs work in other contexts, including AAV and lentiviral vectors, and with additional payloads. Finally, optimized 5′ UTRs need to be ultimately validated in vivo for clinical translation, and translational efficiencies across multiple model systems beyond human cells may need to be taken into account to ensure translatability.

## Methods

**5′ UTR library design and construction**. To select the endogenous sequences, we used publicly available matched RNA-Seq and Ribo-Seq datasets, from three human cell lines/tissues: (i) human embryonic kidney 293T (HEK 293T cells), (ii) human prostate cancer (PC3) cells, and (iii) human muscle tissue. The first two were chosen as these are commonly used cell lines, whereas the third (human muscle tissue) was used because it could be the target tissue of DNA vaccine therapy. We then analyzed the RNA-seq and Ribo-seq datasets and determined their translation efficiency rates and mRNA levels. Per-transcript translation efficiency (TE) was defined as Ribo-seq RPKM/RNA-seq RPKM, where RPKM represents Reads Per Kilobase of transcript per Million mapped reads. Transcripts with insufficient RNA-Seq or Ribo-Seq coverage were discarded. Our final selection of natural 5′ UTRs (Supplementary Data 1) consisted of: (i) top 1505 sequences and bottom 937 sequences from transcripts with highest and lowest TEs from HEK 293T cells, (ii) top 1692 and bottom 756 sequences from transcripts with highest and lowest TEs from PC3 cells, (iii) 1831 sequences from transcripts that displayed maximum TEs for muscle tissue, and (iv) 1693 5′ UTR regions from transcripts with high mRNA expression levels in muscle tissue, which were extracted from publicly available data from the Genotype-Tissue Expression (GTEx) project.

We trained and developed a computational model to predict the TE based on its 5′ UTR characteristics. To establish this model, we first identified which sequence features of 5′ UTRs increased gene expression levels and TE. For this aim, we extracted several sequence features, including k-mer frequency, RNA folding energy, 5′ UTR length, and number of ORFs (The features with highest feature importance are listed in Supplementary Fig. 1). We developed a computational model trained on sequence features to predict TE and mRNA expression under various cell conditions. The model used trained data on experimentally determined translation efficiency rates and mRNA levels, which were obtained from analyzing publicly available RNA-seq and Ribo-seq data of endogenous genes from three human cell types (HEK 293T cells, PC3 cells, and human muscle tissue). We used the *randomforest* R package with default parameters, both to build and to evaluate the model. Data were subdivided for training/testing: 80% of the data was used for training the model and 20% for testing. This process was repeated 5 times for 5 randomly selected non-overlapping test sets (i.e., 5-fold cross validation). The final evaluation metric was the Spearman correlation between predicted translation efficiency and actual translation efficiency (RPKM RiboSeq/RPKM RNAseq).

The workflow consisted of the following steps: (i) extract the sequences features of the 5′ UTRs, including those nucleotides surrounding the AUG region, i.e., the whole 5′ UTR plus 15 bp of the CDS sequences, for each of the expressed transcripts in each cell line or tissue. (ii) Train a Random Forest machine learning method for each cell type/tissue, to learn a function that maps sequence features to mRNA expression and TE. (iii) Design a set of 100 bp synthetic sequences that maximize TE and protein expression (where protein expression is computed as RNA levels * TE). A genetic algorithm (GA) was applied to search the 100-bp sequence space. For each GA run, we randomly sampled 100 endogenous 5′ UTR sequences as an "initial population" prior to undergoing evolution and selected them or their offspring based on their fitness, which is defined by their predicted TE or predicted protein expression from the previous trained model. (iv) From our GA results, we kept the top 5 sequences with at least 5 bp differences in each run. (v) To validate the accuracy of our model, we also selected sequences with a small number of mutations from the sequence's endogenous origin, but large increase/decrease of TE or Protein RNA expression compared to that of the endogenous sequence. The first set was designed by evolving 1198 5′ UTRs within 2 generations to test the algorithm (Supplementary Data 2). The second set of 2388 5′ UTRs was designed as follows: from the output of each run, we took the best five sequences during the run, requiring them to have at least five mismatches of each other, and also requiring that their scores were at least 0.05 better than those of the initial naturally occurring sequences within a maximum of 50 generations (Supplementary Data 3).

A 12 K oligonucleotide library of 140-mer was synthesized using CustomArray to contain 100 bp variable 5′ end sequences flanked by PCR priming sites. The ssDNA of each 140 bp consisted of two 20-bp homologous regions at two ends and 100 bp representing the 5′ UTR (from the 5′ UTR library). The details of the library are in Supplementary Data 4. The library was cloned to the reporter plasmids using conventional restriction enzyme cloning and Gibson Assembly.

**Plasmid construction**. The plasmids used in this study were built using restriction enzyme cloning and Gibson assembly. The primers used in this study are in

Supplementary Table 1. The plasmid maps are available in FigShare (https://doi.org/10.6084/m9.figshare.14624472.v1).

**The construction of the HEK 293T landing pad cell lines**. HEK 293T cells (ATCC, VA, USA) were grown in polystyrene flasks in Dulbecco's Modified Eagle's Medium (Life Technologies, CA, USA) supplemented with 10% fetal bovine serum (VWR, PA, USA) and 1% penicillin/streptomycin (Life Technologies, CA, USA) at 37 °C and 5% $CO_2$. When the cells were 80–90% confluent, cells were harvested with 0.25% trypsin (Life Technologies, CA) for transfection. To make lentivirus containing the landing pad, HEK 293T cells were plated in 6-well plate format. In brief, 12 μL of FuGENE HD (Promega, WI, USA) was mixed with 100 μL of Opti-MEM medium Life Technologies, CA) and was added to a mixture of the three plasmids: 0.5 μg of lentiviral envelop vector pCMV-VSV-G vector, 0.5 μg of lentiviral packaging vector psPAX2, and 1 μg of lentiviral expression vector for landing pad insertion pJC191 (Supplementary Fig. 9). After 20 min incubation of FuGENE HD/DNA complexes at room temperature, 1.8 million cells were added to each FuGENE HD/DNA complex tube, mixed well, and incubated for another 10 min at room temperature before being added to 6-well plates containing 1 mL cell culture medium, followed by incubation at 37 °C and 5% $CO_2$. The media was removed 24 h after transfection and 2 mL fresh media was added. After another 24 h transfection, supernatant containing newly produced viruses was collected, and filtered through a 0.45 mm syringe filter (Pall Corporation, MI, USA) and used for infection. The filtered supernatant was diluted by different titrations of viruses using fresh media, and mixed with 8 mg/mL polybrene before added into 6-well plates with 1 million cells seeded on each well 24 h before infection. Cell culture medium was replaced the next day after infection and cells were cultured for at least 3 days prior to FACS analysis or sorting using BD FACSAria. Single YFP positive cells from the well with less than 10% YFP positive cells (roughly 0.1 MOI) were sorted into a 96-well plate and were cultured in fresh medium for 2 weeks and expanded to 6-well plates with medium supplemented with 50 μg/mL hygromycin (Life Technologies, CA, USA). Two clones with single copy landing pad insertion, HEK-LP3, and HEK-LP9, were selected as the parental landing pad cell lines for library screening.

**Library transfection, recombinase-based library integration, and next-generation sequencing**. The landing pad cells were seeded as 1 million per well on 6-well plates 24 h before transfection. One μg library plasmid pJC253L (Supplementary Fig. 10) carrying an attB site and the 5′ UTR library and 1 μg BxbI recombinase expressing plasmid pCAG-BxbI were mixed with 6 μL FuGENE HD and added into each well. Eight wells of each landing pad cells were used for transfection. To ensure the reproducibility of our screening results, we maintained >25-fold coverage of each library member throughout the screening pipeline. 4 μg/mL puromycin was added three days post-transfection, and the cells were cultured for at least one more week. The cells were then analyzed using FACS and sorted into three bins based on distinct levels of GFP intensity while the unsorted cells were used as control. We chose three sorting brackets of top 0–2.5%, 2.5–5%, and 5–10%, instead of only one bracket of 0–10%, to reduce the false positive candidates. Only the candidates that were significantly over-represented in all three bins, relative to background (0–100% bin), were selected as candidates for further validation.

For NGS library preparation, the genomic DNA was extracted from each bin and 800 ng were used as the template for PCR amplification with barcoded Pi7 primer. Sequencing was performed at the MIT BioMicro Center facilities on an Illumina NextSeq machine using 150 bp double-end reads.

**Lentiviral-based screening of the 5′ UTR library in HEK 293T cells**. When HEK 293T cells were 80–90% confluent, cells were harvested with 0.25% trypsin for transfection. For each well, 12 μL of FuGENE HD (Promega, WI, USA) was mixed with 100 μL of Opti-MEM medium Life Technologies, CA, USA) and added to a mixture of the three plasmids: 0.5 μg of lentiviral envelop vector pCMV-VSV-G vector, 0.5 μg of lentiviral packaging vector psPAX2, and 1 μg of lentiviral 5′ UTR library plasmid pJC240L (Supplementary Fig. 11). After 20 min incubation of FuGENE HD/DNA complexes at room temperature, 1.8 million cells were added to each FuGENE HD/DNA complex tube, mixed well, and incubated for another 10 min at room temperature before being added to 6-well plates containing 1 mL cell culture medium, followed by incubation at 37 °C and 5% CO2. The media was removed 24 h after transfection and 2 mL fresh media was added. After another 24 h transfection, supernatant containing newly produced viruses was collected, filtered through a 0.45 mm syringe filter, and used for infection.

The filtered supernatant was diluted by titration of viruses using fresh media, and mixed with 8 mg/mL polybrene before added into 6-well plates with 1 million HEK 293T cells seeded on each well 24 h before infection. Cell culture medium was replaced the next day after infection and the infection efficiency was calculated for at least 3 days prior to FACS analysis using BD LSR II. The infected HEK 293T cells from the well with less than 10% GFP positive cells (roughly 0.1 MOI) were selected, as the integration of a single copy of the 5′ UTR was expected in most of the infected cells. To ensure the reproducibility of the screening results, we maintained >25-fold coverage of each library member throughout the screening pipeline. The infected cells were sorted and further expanded for at least one week.

The cells were then analyzed using FACS and sorted into three bins based on distinct levels of GFP intensity while the unsorted cells were used as control.

For NGS library preparation, the genomic DNA was extracted from each bin and 800 ng were used as the template for PCR amplification with barcoded Pi7 primer. Sequencing was performed at the MIT BioMicro Center facilities on an Illumina NextSeq machine using 150 bp double-end reads.

**NGS data pre-processing and analysis**. Fastq files were first inspected for quality control (QC) using FastQC. Fastq files were then filtered and trimmed using fastx_clipper of the FASTX-Toolkit. Trimmed fastq files were collapsed using fastx_collapser of the FASTX-Toolkit. The collapsed fasta file was used as an input for alignment in Bowtie2 with a very sensitive alignment mode and aligned against the library reference. The resulting SAM file was filtered for mapped reads using SAMtools, and the reads were then quantified by summing the counts of each unique promoter using an in-house R script. The reads were normalized by dividing all reads in the sample by a size factor estimated by DESeq2. Replicability was assessed using Pearson correlation values using the cor.test function R. Differentially expressed 5′UTRs were identified using DESeq2. Top-ranked 5′ UTRs, which were ranked based on their log2 fold change relative to the unsorted bin, were selected as *leads* for experimental validation.

**Measurement of the GFP expression of the plasmids with 5′ UTR candidates in mammalian cells**. HEK 293T cells, human rhabdomyosarcoma (RD) cells, human breast adenocarcinoma (MCF-7) cells, and mouse C3H muscle myoblast (C2C12) cells were obtained from the American Type Culture Collection. HEK-293T, RD, MCF-7, and C2C12 cells were cultured in DMEM supplemented with 10% fetal bovine serum and 1% Pen/Strep at 37 °C with 5% $CO_2$. When the cells were 80–90% confluent, cells were harvested with 0.25% trypsin for transfection.

For HEK 293T cells, 50,000 cells per well on 96-well plates were plated 24 h before the transfection and 50 ng of plasmid with different 5′ UTRs or introns (Supplementary Tables 2 and 3) and 50 ng pEF1α-BFP was mixed with 0.3 µL FuGENE HD used in each well; for RD cells, 10,000 cells per well on 96-well plates were plated 24 h before transfection and 50 ng of plasmid with pJC271 (Supplementary Figs. 12 and 13) or plasmids with different 5′ UTRs and 50 ng pEF1α-BFP was mixed with 0.5 µL FuGENE HD used in each well; for MCF-7 cells, 20,000 cells per well on 96-well plates were plated 24 h before transfection and 50 ng of plasmid with different 5′ UTRs and 50 ng pEF1α-BFP was mixed with 0.5 µL FuGENE HD used in each well; for C2C12 cells, 10,000 cells per well on 96-well plates were plated 24 h before transfection and 50 ng of plasmid with different 5′ UTRs and 50 ng pEF1α-BFP was mixed with 0.3 uL Lipofectamine 2000 (Life Technologies, CA, USA) used in each well. After one or two days, the GFP and BFP intensity were measured using BD LSR II (Supplementary Fig. 14).

**ELISA measurement of therapeutic protein production**. To determine production of therapeutic proteins with 5′ UTR candidates, we constructed plasmids encoding secretory human vascular endothelial growth factor (hVEGF) or C–C Motif Chemokine Ligand 21 (hCCL21), downstream of different 5′ UTR candidates, respectively. HEK 293T cells were transfected with 100 ng of plasmid in 24-well plates at 100,000 cells per well and cultured for 24 h with l mL complete culture medium. After washing cells once with PBS, complete culture medium was replaced with 0.5 mL plain DMEM supplemented with 1% Pen/Strep. Cells were incubated at 37 °C with 5% CO2 for an additional 24 h. Supernatants were then collected, spun down at 350 × g, and stored at −80 °C. The amount of each human protein in the supernatant was quantified by enzyme-linked immunosorbent assay (ELISA). Concisely, hVEGF concentration was determined by human VEGF ELISA Kit (KHG0111, Thermo Fisher Scientific), following the manufacturer's instructions; hCCL21 concentration was determined by human CCL21/6Ckine DuoSet ELISA (DY366, R&D systems), following the manufacturer's instructions. Data are presented as pg/ml per 100,000 cells per 24 h.

**Statistical analysis**. All quantitative data are presented as mean ± standard deviation (SD). The statistical analyses were performed with GraphPad Prism 7.0 (GraphPad Software, La Jolla, CA, USA) statistics software.

**Reporting summary**. Further information on research design is available in the Nature Research Reporting Summary linked to this article.

## Data availability

Data supporting this study are presented in the main text and Supplementary information, and are available from the corresponding authors upon request. The plasmid maps are available in FigShare (https://doi.org/10.6084/m9.figshare.14624472.v1). Raw FASTQ data has been deposited in the Gene Expression Omnibus, under the accession code GSE176581. Ribosome profiling data used in this work corresponds to GSE55195 (HEK293T), GSE35469 (PC3), and GSE56148 (muscle). This work used public RNAseq data from the GTEx database (https://gtexportal.org/). The processed data (extracted 5′ UTR features) used for model training has been made publicly available in GitHub (https://github.com/zzz2010/5UTR_Optimizer). Source data are provided with this paper.

## Code availability

All code used to analyze, design, evolve, and select optimized 5′ UTR sequences for enhanced protein expression is publicly available in GitHub (https://github.com/zzz2010/5UTR_Optimizer). A stable release of the 5UTR_Optimizer code (version 1.0) is available at Zenodo (https://doi.org/10.5281/zenodo.4782661).

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

## Acknowledgements

The authors thank Dr. Stuart Levine at MIT MicroBio Center for assisting with NGS, and Karen Pepper for help with paper editing. The work was financially supported by Army Research Office (funding under the OSP account 6924758), Boston University (funding under the OSP account 6924758), HFSP to T.K.L., and NIH (R01 GM113708, R01 HG004037) to M.K.. E.M.N. thanks Human Frontier Science Program (LT000307/2013-L) for their financial support.

## Author contributions

J.C., E.M.N., G.C.G.C., M.K., and T.K.L. conceived idea and designed the study. E.M.N., Z.Z., and G.C.G.C. designed the 5' UTR library. J.C. designed and performed experiments and analyzed data. E.M.N. performed the NGS data analysis. Z.Z. developed the machine learning approaches. J.C., E.M.N., Z.Z., and D.L. performed the computational analysis. W.C. performed the ELISA experiments. G.C.G.C. and A.S.L.W. helped with library cloning. J.C., E.M.N., Z.Z., G.C.G.C., M.K., and T.K.L. wrote the paper. All authors discussed the results and edited the manuscript.

## Competing interests

J.C., E.M.N., Z.Z., M.K., and T.K.L. have filed patent applications on the work. The applicant is Massachusetts Institute of Technology. The application number:16/441,647. T.K.L. is a co-founder of Senti Biosciences, Synlogic, Engine Biosciences, Tango Therapeutics, Corvium, BiomX, Eligo Biosciences, Bota.Bio, and Avendesora. T.K.L. also holds financial interests in nest.bio, Ampliphi, IndieBio, MedicusTek, Quark Biosciences, Personal Genomics, Thryve, Lexent Bio, MitoLab, Vulcan, Serotiny, and Avendesora. The other authors declare no competing interests.
