## [Peer Review File · Nature Communications]

Reviewers' Comments:

Reviewer #1:

Remarks to the Author:

The manuscript "High-Throughput 5' UTR Engineering for Enhanced Protein Production in Non-Viral Gene Therapies" by Cao et al., presents high-throughput strategy to design, screen, and optimize novel 5'UTRs to enhance protein expression. While study brings an interesting concept in the design of 5'UTRs the data as well as author's claims are unsubstantiated with currently presented results. The manuscript does not provide enough analyses and data to support claims of the authors. In a current form it does not represent anything more than a collection of 12 000 sequences that were tested for the GFP expression. It does not provide enough evidence that these 5'UTR sequences either increase translational efficiency (TE) or RNA stability. Analyses of "natural" 5'UTRs in Ribo-Seq data does not take in account that majority of human 5'UTRs are longer than 100bp and that multiple genes have 5'UTR isoforms with different contributions to overall TE and RNA stability of these genes. Authors do not take this in account. Even if the argument of same transcription rate stands, which is questionable, authors do not provide any data on RNA stability or association of these 5'UTRs with 40S subunit of ribosome (translation initiation efficiency).

Manuscript also does not represent enough novelty in the method approach given that previous studies Ref 19 (Sample et al., Nature Biotechnology, 2019) have already constructed similar libraries.

Analyses of polysome associated RNAs (40S, monosome, polysome peaks separated) and RNASeq of the library in the tested cells would answer some of the questions. The reasoning behind 100bp 5'UTR is lacking and GFP FACS sorting brackets (levels of GFP expression) are missing. There are crucial questions that require further mechanistic explanation and support for the claims in increase in translation efficiency.

As such, my suggestion is that manuscript is not of enough quality for Nature communications.

Reviewer #2:

Remarks to the Author:

SUMMARY

In "High-Throughput 5' UTR Engineering for Enhanced Protein Production in Non-Viral Gene Therapies" (manuscript 250914) the authors develop a high-throughput strategy to design, screen, and optimize novel 5' UTRs that enhance protein expression. Current gene therapies, although promising, still face many issues with potency, cost, and immunogenicity. Here, the authors improve the protein expression in a non-viral gene therapy by optimizing the 5' UTR of the nucleic acid payload. In their high-throughput strategy, the authors used genetic algorithms to generate synthetic 5' UTRs from known naturally occurring 5' UTRs, and developed a recombinase-based library screening strategy for screening. This work resulted in three synthetic 5' UTRs that outperform commonly used non-viral gene therapy plasmids to enhance protein expression across a variety of cell types.

GENERAL ASSESSMENT

Overall, this study is technically sound and should interest a range of researchers in the genetic engineering, synthetic/systems biology, and gene therapy fields. The concept of engineering the 5' UTR to improve nucleic acid payload in gene therapy has only been minimally explored, and the methodology presented could in principle be applied to other regulatory elements. Furthermore, the authors discover and report specific 5' UTR sequences that researchers could use to increase protein expression for various applications.

There are four main issues that need to be addressed, which will then make this manuscript suitable for publication (listed in more detail below). First, the authors should provide a more in-depth discussion of prior work to better contextualize their study and its contributions. Second, the

specifics of the machine learning algorithms used, both the Random Forest Classifier and the Genetic Algorithm, need to be better described: specifically, the rationale for choosing the specified features, how the RFC was trained and tested, and how the results of the RFC are used to inform the genetic algorithm. Third, the authors should provide more analysis about the results of their 5' UTR screening. Fourth, the authors should clarify the role of predicting mRNA levels in their workflow.

MAJOR CRITIQUES

1. I recommend the authors provide more context of prior work done on 5' UTR engineering for improving translation efficiency. Although the authors cited a few examples (ref. 24-27) and claim that knowledge is limited and UTR engineering remains challenging (all true!), the specific contributions and limitations of these works should be described to help readers better understand why the current approach might overcome such challenges. In particular, the authors should discuss:

a. Ref. 27. (Sample, P., et. al)

Conceptually similar. Sample et al. has much better performance characterization of the models. Sample et al. focused on protein expression from mRNA directly, which seems to yield more generalizable sequence-function insights, whereas the present study focused on a very specific expression scenario (expressed from recombinase integrated constructs). Are similar sequence-function relationships found?

b. Ref. 24-26.

Although these studies were focused on yeast UTR engineering, the general approaches are conceptually similar, where predictive models were built and high throughput reporter assays used.

c. "Modular 5'-UTR hexamers for context-independent tuning of protein expression in eukaryotes", *Nucleic Acids Research*, Volume 46, Issue 21, 30 November 2018, Page e127. This citation would offer the authors an opportunity to highlight advantages of machine learning approaches over just screening randomized libraries.

2. The authors used 5' UTR sequence features to train a random forest regression model to predict TE and mRNA expression. I believe that some further data, analysis, and explanation is needed:

a. The authors do not describe how they partitioned the data set into training and test data.

b. What were the performance metrics for each random forest model?

c. What were the most important features for each model? Are they the same across data sets?

d. The authors do not describe how the output of the random forest classifier models are used to inform the design of synthetic 5' UTRs using a genetic algorithm (unclear in the code)

e. The random forest classifier uses "5' UTR length" as a sequence feature to train the model.

However, all of the synthetic 5' UTRs are designed to be 100 bases. If this is the case, then how does the feature importance of sequence length inform the design of the synthetic 5' UTRs?

f. The random forest classifier uses "codon usage" as a sequence feature to train the model.

However, (1) why would codon usage affect UTR function, since codons are outside of UTR and (2) it is not clear how this is incorporated into the design of synthetic 5' UTRs?

3. The authors found 7 high expression UTRs from the 8414 natural UTRs. Are these 7 UTRs from the High TE groups? What is the distribution of High TE and Low TE UTRs in the screening result? What is the agreement between the library and the FACS screening?

4. What is the motivation and reason for predicting mRNA levels, and how are the results from those machine learning models used for 5' UTR engineering?

MINOR CRITIQUES

5. Figure 1. In the text, Figure 1 is referenced as describing the in silico generation of the synthetic 5' UTRs. However, Figure 1 is a high-level diagram of the entire workflow with no details about the in silico generation of the 5' UTRs. Please better cite the figure in the text.

6. Figure 2: This figure can be refined further.

- a. Are the data sets all the same size, as suggested by the donut plot? If not, the sizes should be adjusted.
- b. What is the RPKM filter and is it the same for all data sets?
- c. The section labelled "model evaluation" seems more like a feature selection step
- d. Where is the arrow labelled "training target input" originating from, and what it is trying to convey?
- e. How does the output of the random forest model inform the genetic algorithms?

7. NGS Data Set

- a. In the methods, the authors state that "1831 sequences from transcripts that displayed maximum TEs for muscle tissue" are selected, however in Figure 2, it suggests that transcripts with "variable TEs" were selected. Please clarify which is correct, and also explain why only transcripts with maximum TEs were selected, when the other data sets used high and low sequences.
- b. What filters and or cutoffs were used for the RNA-Seq and Ribo-Seq data sets to yield the "high" and "low" sets of 5' UTR sequences (e.g. top 1505 sequences and bottom 937 sequences from HEK293)
- c. Please explain why the Low TE UTRs are included in the final library for screening.

8. NGS Data Analysis

- a. The authors do not mention which reference genome was used for FASTQ sequence alignment.
- b. For feature extraction, the authors use features from the UCSC Hg19 genome build. Why was this build chosen over the more up-to-date Hg38 genome build?
- c. What is the RPKM filter (see Figure 1) and is it the same for all data sets?

9. The authors were able to isolate several high expression UTRs, but did not discuss why these particular sequences lead to high expression. From the machine learning and library screening, did the authors find any sequence features that lead to high expression? Is there any new knowledge generated?

10. After library integration by recombinase, the cells were grown for one week and selected for 4 days according to the main text. However according to the methods they were grown for 3 days and then selected for one week. Please clarify.

11. For FACS screening of the recombinase library, have the authors confirmed that transfected plasmids were completely degraded at the time of sorting?

12. Supplementary Code

- a. Two scripts are labelled "5" (no #4).
- b. There is no difference between scripts 6 and 7 (other than the title and some syntax not affecting the code). Please clarify.
- c. As written, the code could not be used by a new user due to many components being hardcoded specific to this application. The impact of this work could be enhanced if the code is packaged into a more streamlined workflow.

ANSWERS TO REVIEWER'S COMMENTS

Reviewer #1 (Remarks to the Author):

The manuscript "High-Throughput 5' UTR Engineering for Enhanced Protein Production in Non-Viral Gene Therapies" by Cao et al., presents high-throughput strategy to design, screen, and optimize novel 5'UTRs to enhance protein expression. While study brings an interesting concept in the design of 5'UTRs the data as well as author's claims are unsubstantiated with currently presented results. The manuscript does not provide enough analyses and data to support claims of the authors. In a current form it does not represent anything more than a collection of 12 000 sequences that were tested for the GFP expression. It does not provide enough evidence that these 5'UTR sequences either increase translational efficiency (TE) or RNA stability.

We thank the reviewer for his/her comments. To address this issue, we have edited the document to highlight that our screen focuses on identifying UTRs that enhance gene expression, rather than emphasizing increased TE or RNA stability. In Lines 83-84, the revised sentences now read: "In this study, we developed a platform to systematically screen and engineer novel 5' UTRs that can enhance protein expression in mammalian cells (**Fig. 1**)."

In Lines 265-270, we inserted the sentences "Compared with elucidating mechanisms for validated 5' UTRs, such as the key role of Kozak sequences²⁷ or harpins near the 5' cap^{60,61}, our work instead focus on identifying novel 5' UTRs that are stronger than that on the commercially and clinically used gene therapy vectors and the commonly used introns and other 5' UTRs. Future work could include investigations into the underlying mechanisms that enhance gene expression from these 5' UTRs to provide insights for further improvements."

Deciphering whether 5' UTRs increase TE, RNA stability, or both is not the main goal of our work. Previous work, such as that by Sample *et al.*, Nature Biotech 2019, used ribosome profiling as a readout to precisely examine how 5' UTR diversity affects translational efficiency, and to then build a model that would predict TE based on 5' UTR features. By contrast, here we provide a distinct and novel strategy to design, screen, diversify and finally optimize synthetic 5' UTRs to obtain increased protein expression levels, and show how to refine them combinatorially to maximize expression levels. The main goal of our work is thus to develop a platform that can discover and engineer 5' UTRs that enhance gene expression in mammalian cells for vaccine production and other biomedical applications, in order to fill a significant unmet need and challenge in the fields of gene therapy and drug manufacturing.

Analyses of "natural" 5'UTRs in Ribo-Seq data does not take in account that majority of human 5'UTRs are longer than 100bp and that multiple genes have 5'UTR isoforms with different contributions to overall TE and RNA stability of these genes. Authors do not take this in account.

The reviewer is correct that natural 5'UTRs are sometimes longer than 100bp; however, this is a constraint due to DNA synthesis limits that restricts our ability to make significantly larger artificial sequences. The same limitation has been encountered by others using massive parallel reporter 5' UTR assays, such as Sample et al., Nature Biotech 2019, who actually used 50bp long 5' UTRs. The reviewer is also correct in stating that there are some examples of human genes that have common 5' UTR isoforms and yet their TE might not be identical, and this will again be a limitation of the method. We have now discussed these limitations in the Discussion section of our manuscript and thank the reviewer for pointing them out.

In Lines 275-278, we inserted the following sentence "However, we should note that naturally occurring 5' UTRs in mammalian species are often longer than 100 bp, Future work exploring 5' UTR diversity might benefit from improved DNA synthesis technologies that will make it possible to expand the length of the diversity region that is tested in the screening."

Even if the argument of same transcription rate stands, which is questionable, authors do not provide any data on RNA stability or association of these 5'UTRs with 40S subunit of ribosome (translation initiation efficiency).

The reviewer mentions that there is not enough evidence that these 5' UTR sequences will increase translational efficiency or RNA stability and proposes that we analyze polysome associated RNAs in the tested cells. While this is an interesting question, it is not the main question or goal of this work. Here, we do not aim to provide a mechanistic dissection of why specific 5' UTRs perform better than others (which has been addressed by other work, such as the work mentioned by the reviewer, Sample *et al.*, Nature Biotech 2019). Rather, here we aim to provide a novel framework to design, screen, and optimize libraries to enhance expression levels using non-viral gene therapies, which is the goal of our project. Importantly, we show that our approach succeeds at achieving this goal, showing that some synthetic and combinatorial 5' UTRs increase protein expression levels up to 400%. We have edited the manuscript to remove any direct claims that we are enhancing RNA stability or translational initiation and focus on the improvement of gene expression only. In Lines 83-84, the revised sentence now reads: "In this study, we developed a platform to systematically screen and engineer novel 5' UTRs that can enhance protein expression in mammalian cells (**Fig. 1**)."

Manuscript also does not represent enough novelty in the method approach given that previous studies Ref 19 (Sample *et al.*, Nature Biotechnology, 2019) have already constructed similar libraries.

While there are some similarities to the work mentioned above (Sample *et al.*, Nature Biotechnology 2019), our work differs in several major key points:

- 1) The main goal of our work is to develop a platform that allows us to discover and engineer 5' UTRs that enhance gene expression in mammalian cells for gene therapy. By contrast, the work of Sample *et al.* is mainly focused on developing novel models that can predict translational efficiency of 5' UTRs.
- 2) The work by Sample *et al.* did not discover or engineer 5' UTRs that outperformed the initial 5' UTRs that were tested in their experiment. In this regard, while expression levels can be tuned down by adding hairpins to 5' UTRs, the unsolved issue is how to upregulate gene expression using 5' UTRs, which is the main focus of our work.
- 3) Our work includes the development of a recombinase-based library screening approach, which eliminates the copy number and position effects in conventional lentiviral-based library screening. To our knowledge, our study is the first time that this approach has been used in library screening, and we show its utility in enhancing the reproducibility of screening. We envision that our approach will become a popular alternative library screening method and, particularly, an alternative to conventional lentiviral-based library screening.
- 4) In our work, we demonstrate the broad applicability of these synthetic 5' UTRs in multiple mammalian cell lines to increase the expression of therapeutic proteins, so that these 5' UTRs can be added to the toolbox of 5' UTRs/introns for gene expression or protein production. In contrast, Sample *et al.* tested their 5' UTRs by using fluorescent proteins only in HEK 293T cells. The applicability to diverse cell lines/cell types is a key point that was not tackled in the work from Sample *et al.*
- 5) Sample *et al.* explored smaller regions of diversity (50 bp), and these were generated randomly (Figure 1B from Sample *et al.*). By contrast, we explored 100bp oligos, and these included endogenous and machine-learning evolved sequences, not randomized sequences.
- 6) Sample *et al.* oligos included the Kozak sequence in the randomized region (i.e., they explored diversity in the region that includes the Kozak sequence), and thus, as could be expected, they found that 5' UTRs

that include strong Kozak sequences led to higher ribosome recruitment, whereas 5' UTRs that include weak Kozak sequences were found to lead to weaker ribosome recruitment (Figure 1D,E in Sample et al.). Thus, they obtained very good sequence-function relationships because they included the Kozak sequence within the region explored as “variable”. The Kozak sequence has been previously reported in literature to be a major feature defining 5' UTR strength in mammalian systems, so it is not surprising that this is what they found. Rather than exploring the Kozak sequence as a diversity region, we explored whether variability in the 5' UTR region upstream of the Kozak sequence (which is kept constant and strong in all our plasmids) can lead to enhanced protein expression levels. Indeed, in all our constructs we used a strong and constant Kozak sequence (because pVAX already has a very strong Kozak sequence, and we want to obtain plasmids that outperform pVAX for protein production). We have now explained these differences and discussed our findings in the context of this previous work in the Discussion section.

In Lines 262-283, we inserted the following sentences “Previous work has demonstrated that machine learning methods can be employed to predict 5' UTR translation efficiency in mammalian cells and yeast²⁴⁻²⁷. Here, we extended the use of machine learning from prediction to the de novo design of novel 5' UTRs that would potentially enhance protein expression in human cells, which imposes a key challenge in gene therapy and drug manufacturing. Compared with elucidating mechanisms for validated 5' UTRs, such as the key role of Kozak sequences²⁷ or harpins near the 5' cap^{60,61}, our work instead focus on identifying novel 5' UTRs that are stronger than that on the commercially and clinically used gene therapy vectors and the commonly used introns and other 5' UTRs. Future work could include investigations into the underlying mechanisms that enhance gene expression from these 5' UTRs to provide insights for further improvements.

Lentiviral-based library screening is often a common choice for massive parallel reporter assays. Here, we observed that lentiviral-based 5' UTR screenings yielded poor reproducibility across biological replicates (**Fig. 3C**). To overcome this limitation, we constructed a novel recombinase-based library screening platform, which we find eliminates the copy number and position effects in conventional lentiviral-based screening, and is virtually applicable for screening of any genetic element of interest. Our library of 12,000 5' UTRs was originally synthesized in the form of an array of 100-bp long DNA oligonucleotides and subsequently cloned into vectors (**Fig. 1**). However, we should note that naturally occurring 5' UTRs in mammalian species are often longer than 100 bp, Future work exploring 5' UTR diversity might benefit from improved DNA synthesis technologies that will make it possible to expand the length of the diversity region that is tested in the screening. In this regard, here we found that individual synthetic 5' UTRs can be combined to enhance protein expression. In the future, these regions could be combined with novel translation initiation site (TIS) sequences⁶² that might potentially outperform the commonly used Kozak sequences for gene therapy.”

Analyses of polysome associated RNAs (40S, monosome, polysome peaks separated) and RNASeq of the library in the tested cells would answer some of the questions. The reasoning behind 100bp 5'UTR is lacking and GFP FACS sorting brackets (levels of GFP expression) are missing. There are crucial questions that require further mechanistic explanation and support for the claims in increase in translation efficiency.

We chose 100 bp because of the limits of commercially available ssDNA template synthesis when we synthesized the library in 2017. The ssDNA of 140 bp consists of two 20-bp homologous regions at two ends and 100 bp 5' UTR library. This information has been now clarified in the Methods section.

We chose three sorting brackets of top 0-2.5%, 2.5-5% and 5-10%, instead of only one bracket of 0-10%, because we wanted to reduce the false positive candidates. Only the candidates that appeared in all three bins relative to background (0-100% bin) were selected as candidates for further validation. This information has now been included in the Methods section.

In Lines 140-143 (Supporting Information), we inserted the following sentences “We chose three sorting brackets of top 0-2.5%, 2.5-5% and 5-10%, instead of only one bracket of 0-10%, to reduce the false positive candidates. Only the candidates that were significantly over-represented in all three bins, relative to background (0-100% bin), were selected as candidates for further validation.”

As such, my suggestion is that manuscript is not of enough quality for Nature communications.

We hope that with these clarifications we have now addressed the reviewer’s concerns, and that the Reviewer will now find our improved work suitable for publication.

Reviewer #2 (Remarks to the Author):

SUMMARY

In “High-Throughput 5’ UTR Engineering for Enhanced Protein Production in Non-Viral Gene Therapies” (manuscript 250914) the authors develop a high-throughput strategy to design, screen, and optimize novel 5’ UTRs that enhance protein expression. Current gene therapies, although promising, still face many issues with potency, cost, and immunogenicity. Here, the authors improve the protein expression in a non-viral gene therapy by optimizing the 5’ UTR of the nucleic acid payload. In their high-throughput strategy, the authors used genetic algorithms to generate synthetic 5’ UTRs from known naturally occurring 5’ UTRs, and developed a recombinase-based library screening strategy for screening. This work resulted in three synthetic 5’ UTRs that outperform commonly used non-viral gene therapy plasmids to enhance protein expression across a variety of cell types.

GENERAL ASSESSMENT

Overall, this study is technically sound and should interest a range of researchers in the genetic engineering, synthetic/systems biology, and gene therapy fields. The concept of engineering the 5’ UTR to improve nucleic acid payload in gene therapy has only been minimally explored, and the methodology presented could in principle be applied to other regulatory elements. Furthermore, the authors discover and report specific 5’ UTR sequences that researchers could use to increase protein expression for various applications.

There are four main issues that need to be addressed, which will then make this manuscript suitable for publication (listed in more detail below). First, the authors should provide a more in-depth discussion of prior work to better contextualize their study and its contributions. Second, the specifics of the machine learning algorithms used, both the Random Forest Classifier and the Genetic Algorithm, need to be better described: specifically, the rationale for choosing the specified features, how the RFC was trained and tested, and how the results of the RFC are used to inform the genetic algorithm. Third, the authors should provide more analysis about the results of their 5’ UTR screening. Fourth, the authors should clarify the role of predicting mRNA levels in their workflow.

We thank the reviewer for his/her comments and for his/her time in reviewing our work. We will address the four main points that the reviewer mentions above in the individual point-by-point answers below.

MAJOR CRITIQUES

1. I recommend the authors provide more context of prior work done on 5’ UTR engineering for improving translation efficiency. Although the authors cited a few examples (ref. 24-27) and claim that knowledge is limited and UTR engineering remains challenging (all true!), the specific contributions and limitations of these works should be described to help readers better understand why the current approach might overcome such challenges. In particular, the authors should discuss:

a. Ref. 27. (Sample, P., et. al). Conceptually similar. Sample et al. has much better performance characterization of the models. Sample et al. focused on protein expression from mRNA directly, which seems to yield more generalizable sequence-function insights, whereas the present study focused on a very specific expression scenario (expressed from recombinase integrated constructs). Are similar sequence-function relationships found?

We thank the reviewer for this comment. As the reviewer mentions, Sample et al. focused on mRNA measurements rather than protein measurements, and developed algorithms that can predict the translational efficiency of certain 5' UTRs. Some key differences between their work and our work are:

- Sample *et al.* used their own generated experimental RiboSeq data to train their machine-learning model. By contrast, we used data from other studies to train our models, which were used, in turn, to evolve the sequences to generate synthetic 5' UTRs, which were then synthesized.
- Sample *et al.* explored smaller regions of diversity (50 bp), and these were generated randomly (Figure 1B from Sample *et al.*). By contrast, we explored 100 bp oligos, and these included endogenous and machine-learning evolved sequences, not randomized sequences.
- Sample *et al.* oligos included the Kozak sequence in the randomized region (i.e., they explored diversity in the region that includes the Kozak sequence); thus, as could be expected, they found that 5' UTRs that include strong Kozak sequences lead to higher ribosome recruitment, whereas 5'UTRs that include weak Kozak sequences lead to weaker ribosome recruitment (see Figure 1D,E in Sample *et al.*) Because they included the Kozak sequence within the region explored as “variable,” they obtained very good sequence-function relationships. The Kozak sequence has been reported as a major feature defining 5'UTR strength, so what they found is not surprising. By contrast, we were not interested in exploring the Kozak sequence as a diversity region, and we used a strong and constant Kozak sequence in all our constructs. The Kozak sequence we used is that of pVAX: our use of this Kozak sequence thus facilitated comparison with pVAX, as we wanted to obtain plasmids that outperform pVAX for protein production. Thus, what we are exploring here, is whether variability in the 5' UTR region BEFORE the Kozak sequence (which is kept constant and strong in all our plasmids) can lead to enhanced protein expression levels.

Consequently, to address the reviewer's question, we do not find similar sequence-function relationships, mainly because the regions of diversity did not fully overlap, and the main sequence-function relationship found by Sample et al. was the Kozak sequence. In our design, we keep this region constant. We have now added a more detailed description and comparison of our work in the context of their work, in both the Introduction (Page 4) and Discussion (Pages 9-10) sections.

In Lines 76-82, we inserted the following sentences “For example, Sample *et al.*²⁷ screened a library of 50-bp random 5' UTRs including the Kozak sequence in the randomized region, and found that 5' UTRs that include strong Kozak sequences lead to higher ribosome recruitment. However, plasmids commonly used for non-viral gene therapy, such as pVAX1, already contain strong Kozak sequences. Thus, we were not interested in exploring the region that comprised the Kozak sequence as a diversity region but, rather, whether variability in the 5' UTR region preceding a strong Kozak sequence could lead to the identification of novel 5' UTRs that would enhance protein expression levels.”

In Lines 262-270, we inserted the following sentences “Previous work has demonstrated that machine learning methods can be employed to predict 5' UTR translation efficiency in mammalian cells and yeast²⁴⁻²⁷. Here, we extended the use of machine learning from prediction to the *de novo* design of novel 5' UTRs that would potentially enhance protein expression in human cells, which

imposes a key challenge in gene therapy and drug manufacturing. Compared with elucidating mechanisms for validated 5' UTRs, such as the key role of Kozak sequences²⁷ or harpins near the 5' cap^{60,61}, our work instead focus on identifying novel 5' UTRs that are stronger than that on the commercially and clinically used gene therapy vectors and the commonly used introns and other 5' UTRs. Future work could include investigations into the underlying mechanisms that enhance gene expression from these 5' UTRs to provide insights for further improvements.”

b. Ref. 24-26. Although these studies were focused on yeast UTR engineering, the general approaches are conceptually similar, where predictive models were built and high throughput reporter assays used.

The predictive models are conceptually similar; however, we used cell-type specific mRNA/Ribo-seq to build our predictive model. We also used a mammalian system rather than a yeast system. We have now cited these works in our manuscript and discussed them in the Discussion section (Page 9) accordingly.

In Lines 262-270, we inserted the following sentences “Previous work has demonstrated that machine learning methods can be employed to predict 5' UTR translation efficiency in mammalian cells and yeast²⁴⁻²⁷. Here, we extended the use of machine learning from prediction to the *de novo* design of novel 5' UTRs that would potentially enhance protein expression in human cells, which imposes a key challenge in gene therapy and drug manufacturing. Compared with elucidating mechanisms for validated 5' UTRs, such as the key role of Kozak sequences²⁷ or harpins near the 5' cap^{60,61}, our work instead focus on identifying novel 5' UTRs that are stronger than that on the commercially and clinically used gene therapy vectors and the commonly used introns and other 5' UTRs. Future work could include investigations into the underlying mechanisms that enhance gene expression from these 5' UTRs to provide insights for further improvements.”

c. “Modular 5'-UTR hexamers for context-independent tuning of protein expression in eukaryotes”, *Nucleic Acids Research*, Volume 46, Issue 21, 30 November 2018, Page e127. This citation would offer the authors an opportunity to highlight advantages of machine learning approaches over just screening randomized libraries.

We thank the reviewer for this suggestion. We have added this reference accordingly and have included a new paragraph discussing the potential of integrating engineered 5' UTRs and TIS (Kozak sequence) in the Discussion section (Page 10).

In Lines 281-283, we inserted the following sentences “In the future, these regions could be combined with novel translation initiation site (TIS) sequences⁶² that might potentially outperform the commonly used Kozak sequences for gene therapy.”

2. The authors used 5' UTR sequence features to train a random forest regression model to predict TE and mRNA expression. I believe that some further data, analysis, and explanation is needed:

a. The authors do not describe how they partitioned the data set into training and test data.

We thank the reviewer for this comment. We used the *randomforest* R package with default parameters, both to build and to evaluate the model. Data was subdivided for training/testing, where 80% of the data was used for training the model and 20% for testing. This process was repeated 5

times for 5 randomly selected non-overlapping test sets (i.e. 5-fold cross validation). This information has now been added to the manuscript in the Methods section.

In Lines 279-281 (Supporting Information), we inserted the following sentences “We used the *randomforest* R package with default parameters, both to build and to evaluate the model. Data was subdivided for training/testing: 80% of the data was used for training the model and 20% for testing. This process was repeated 5 times for 5 randomly selected non-overlapping test sets (i.e., 5-fold cross validation).”

b. What were the performance metrics for each random forest model?

We thank the reviewer for this comment. The final evaluation metric was the Spearman correlation between predicted translation efficiency and actual translation efficiency (RPKM RiboSeq/RPKM RNAseq). This information has now been added to the manuscript in the Methods section.

In Lines 69-71 (Supporting Information), we inserted the following sentences “The final evaluation metric was the Spearman correlation between predicted translation efficiency and actual translation efficiency (RPKM RiboSeq/RPKM RNAseq).”

c. What were the most important features for each model? Are they the same across data sets?

We find that the most important features for all models are RNAfolding energy for the last 50bp of the 5' UTR (considering G-quadruplex and CG percentage) and different k-mer features enriched in different models, as can be shown from the Figure below. These results have now been included in the form of **Supplementary Figure S1**.

d. The authors do not describe how the output of the random forest classifier models are used to inform the design of synthetic 5' UTRs using a genetic algorithm (unclear in the code)

The output of the random forest model is the TE (translation efficiency) for a given 5' UTR sequence, and that becomes the optimization objective in the genetic algorithm. This information has also been added to the Methods section for further clarification for the readers.

In Lines 69-71 (Supporting Information), we inserted the following sentences “The final evaluation metric was the Spearman correlation between predicted translation efficiency and actual translation efficiency (RPKM RiboSeq/RPKM RNAseq).”

e. The random forest classifier uses “5’ UTR length” as a sequence feature to train the model. However, all of the synthetic 5’ UTRs are designed to be 100 bases. If this is the case, then how does the feature importance of sequence length inform the design of the synthetic 5’ UTRs?

The initial feature set includes “5’ UTR length” as a sequence feature because in the training set, consisting of endogenous 5’ UTRs with different lengths, the random forest model can capture the possible impact from the 5’ UTR length. However, after reviewing feature importance in trained models, we also found that “5’ UTR length” is not among the top importance features.

f. The random forest classifier uses “codon usage” as a sequence feature to train the model. However, (1) why would codon usage affect UTR function, since codons are outside of 5’ UTR and (2) it is not clear how this is incorporated into the design of synthetic 5’ UTRs?

Codon usage was included in the beginning as one of the general sequence features, because we hypothesized that the codon usage of the first codons in the translated region (e.g. 15 bp) may capture some information from the 5’ UTR due to co-evolution with the 5’ UTR. However, we found that the codon usage doesn’t show up as a relevant feature in the final built models.

3. The authors found 7 high expression UTRs from the 8414 natural UTRs. Are these 7 UTRs from the High TE groups? What is the distribution of High TE and Low TE UTRs in the screening result? What is the agreement between the library and the FACS screening?

We thank the reviewer for this comment. The 12,000 5’ UTR library consists of two groups, “potential” and “control”. The “potential” group consists of HEK high TEs, PC3 high TEs, Muscle various TEs, Muscle high RNA expression, and synthetic high TEs, while the “control” group consists of HEK low TEs, PC3 low TEs, and synthetic sequences with various predicted TEs. Eleven of the 13 candidates were from the “potential” group. Five of the seven natural 5’ UTRs were from the “potential” group. Six of the six synthetic 5’ UTRs were from the “potential” group.

4. What is the motivation and reason for predicting mRNA levels, and how are the results from those machine learning models used for 5’ UTR engineering?

We employed two types of training data for a given cell type, RNA-seq (mRNA RPKM levels) and Ribo-seq (RPF RPKM Levels). As we are interested in Translation Efficiency (TE)= (Protein Level)/(mRNA Level), we built two predictive models for each type of data, and computed the predicted TE as our later genetic algorithm optimization objective. Theoretically, we could have directly computed the observed TE as our training label using the observed RPF and mRNA levels and have built only one predictive model. However, the data come from different assays, and it is non-trivial to normalize values across genes and across assays; thus, we considered that predicting both levels would be a better option.

MINOR CRITIQUES

5. Figure 1. In the text, Figure 1 is referenced as describing the in silico generation of the synthetic 5’ UTRs. However, Figure 1 is a high-level diagram of the entire workflow with no details about the in silico generation of the 5’ UTRs. Please better cite the figure in the text.

We have made changes in the revised version.

In Lines 83-84, the revised sentences now read: “In this study, we developed a platform to systematically screen and engineer novel 5’ UTRs that can enhance protein expression in mammalian cells (**Fig. 1**).”

6. Figure 2: This figure can be refined further.

a. Are the data sets all the same size, as suggested by the donut plot? If not, the sizes should be adjusted.

The datasets are not identical as a start as they don't have the same coverage, so the number of genes with minimum 1 read per gene will not be the same. However, this donut plot is meant to be a simplistic representation of the datasets that we were using as starting point, which we then filtered by RPKM and by TE. We have now changed the donut plot representation in the Figure to avoid confusion by the readers.

b. What is the RPKM filter and is it the same for all data sets?

Yes, we used an RPKM filter of 10 both for mRNAseq and RiboSeq datasets.

c. The section labelled "model evaluation" seems more like a feature selection step

We are sorry for the confusion. In our algorithm, we do not have an extra feature selection step and random forest models will select/weight the important features during training. We are using 10-fold cross validation to evaluate the random forest model. To avoid confusion, we have now changed the "model evaluation" to "model generation" in the figure.

d. Where is the arrow labelled "training target input" originating from, and what it is trying to convey?

The training target input is the label information (RNA level, TE) used in training random forest models. The arrow representing the label information is extracted from the RNA-seq experiment data and Ribo-seq experiment data. To make it clear, we have now replaced "training target input" for "RNA level/ Ribo/RNA ratio".

e. How does the output of the random forest model inform the genetic algorithms?

The random forest model serves as a fitness function in the genetic algorithm, and indicates how good the synthetic sequence is, based on either predicted TE or predicted RPF level.

7. NGS Data Set

a. In the methods, the authors state that "1831 sequences from transcripts that displayed maximum TEs for muscle tissue" are selected, however in Figure 2, it suggests that transcripts with "variable TEs" were selected. Please clarify which is correct, and also explain why only transcripts with maximum TEs were selected, when the other data sets used high and low sequences.

The muscle dataset had shallow coverage compared to the HEK 293T and PC3 RiboSeq and RNAseq datasets. Thus, all 1831 sequences that passed the RPKM filters (1831 sequences) were included in the 12K library. Due to the shallow coverage, for this dataset we could not select for "top TE" and "bottom TE", but rather we included all the sequences that we find are highly expressed and highly translated in muscle. Thus, for this reason they are "variable TE" (i.e., transcripts with intermediate TE were not discarded in this dataset, whereas they were discarded in the PC3 and HEK 293T datasets).

b. What filters and or cutoffs were used for the RNA-Seq and Ribo-Seq data sets to yield the "high" and "low" sets of 5' UTR sequences (e.g. top 1505 sequences and bottom 937 sequences from HEK293)

From those transcripts that passed the 10RPKM filter for RNAseq and RiboSeq, we kept 5' UTRs with top 20% and bottom 10% TE for the PC3 and HEK 293T datasets. These 5' UTRs were further filtered to remove redundancy across muscle, PC3 and HEK 293T selected datasets. A distribution of the TEs of the selected 5' UTRs is depicted in the image below, in the form of a stripchart. (note: GTEEx and Synthetic have no TE, as they had no RiboSeq data)

c. Please explain why the Low TE UTRs are included in the final library for screening.

Low TE UTRs were included to assess whether the TE was a major driver in GFP protein expression. We wanted to examine whether high TE 5' UTRs would be more enriched in high GFP bins than low TE 5' UTRs.

8. NGS Data Analysis

a. The authors do not mention which reference genome was used for FASTQ sequence alignment.

We used the 12K fasta sequences as reference for the alignment. This dataset is provided as **Supplementary Tables 1-3**.

b. For feature extraction, the authors use features from the UCSC Hg19 genome build. Why was this build chosen over the more up-to-date Hg38 genome build?

This project was started at the time hg19 was more popular and had richer annotation than hg38. We agree that hg38 is a more up-to-date genome build, but our main results, in terms of methodology as well as global results of the synthetic sequence performance, are genome-build independent. Thus, at this point it wouldn't make sense to change the build considering that the library was built using the 5' UTRs from the hg19 genome build.

c. What is the RPKM filter (see Figure 1) and is it the same for all data sets?

We believe the reviewer might be referring to Figure 2 rather than Figure 1. The RPKM filter used was the same for all datasets and was of 10 RPKM for RiboSeq and for RNAseq datasets.

9. The authors were able to isolate several high expression UTRs, but did not discuss why these particular sequences lead to high expression. From the machine learning and library screening, did the authors find any sequence features that lead to high expression? Is there any new knowledge generated?

The main goal of this manuscript is to develop a platform that can discover and engineer 5' UTRs that enhance gene expression in mammalian cells for gene therapy, such 5' UTRs are a significant unmet need in the fields of gene therapy and drug manufacturing. The exact validation of mechanisms for the validated UTRs is outside scope of this paper, but we agree it would be important to try to elucidate them. The reviewer is correct that from the machine learning and library screening, we can examine which features lead to high expression.

To this end, we extracted features from the endogenous sequences that were used to train the model. We find that RNA folding energy (considered along with GC content and Qquadplex) is the most important feature driving TE as per the machine-learning model (as shown in Response to Reviewer #2, point #2c)

Following the reviewer's suggestion, we examined whether there were any specific features that had been previously extracted to train the models (now adding those corresponding to synthetic sequences) and that would explain whether any specific feature is highly correlated with high GFP expression levels. To this end, "5' UTR scores" for each 5' UTR were computed by taking the median enrichment of a given 5' UTR in the top GFP bins (0-10%), relative to the total bin (0-100%). We then compared the 5' UTR scores to the distinct features that were extracted, using Spearman correlations. Through this analysis, we found that the folding energy of the 5' UTR region was the feature that correlated best with our 5' UTR score ($r=0.27$; $p<2.2\cdot 10^{-16}$, see image below). Thus, we find that highly expressed proteins do not have strong folding energies, suggesting that highly structured 5' UTRs might impede obtaining high amounts of protein products, which would suggest that strong RNA structures might be impeding proper ribosome recruitment and/or scanning.

10. After library integration by recombinase, the cells were grown for one week and selected for 4 days according to the main text. However according to the methods they were grown for 3 days and then selected for one week. Please clarify.

Thanks for pointing this out. Puromycin was added 3 days post transfection; thus, the cells were grown for 3 days and then selected for one week. We have now added this clarification in the Results section.

In Lines 159-160, the sentence now reads: “The transfected cells were grown for three days, then subjected to puromycin selection for one week.”

11. For FACS screening of the recombinase library, have the authors confirmed that transfected plasmids were completely degraded at the time of sorting?

We did not test the remaining plasmids. It had been 10 days after transformation, so we assumed that the plasmids and the GFP produced by the plasmids were negligible due to degradation and dilution during cell proliferation. The doubling-time of HEK 293T cells is about 24-36 hours, so the plasmids were diluted by at least 100-1000-fold at the time of sorting.

12. Supplementary Code

a. Two scripts are labelled “5” (no #4).

Thanks for pointing this out. We have relabeled the two scripts to #4 and #5

b. There is no difference between scripts 6 and 7 (other than the title and some syntax not affecting the code). Please clarify.

The only difference is that script 7 uses predicted log Ribo-seq readout as fitness value in GE, which is the sum of predicted log TE and predicted log RNA level.

```
predict_TE<-function(featVec){  
  predict(full.model.te,featVec)+predict(full.model.rna,featVec)}
```

c. As written, the code could not be used by a new user due to many components being hardcoded specific to this application. The impact of this work could be enhanced if the code is packaged into a more streamlined workflow.

We agree that the code could be better structured and packaged for broad use. But due to resource constraints, we cannot achieve this goal in the short term. Our code is publicly available in GitHub (https://github.com/zzz2010/mit_5utr_project), and new users can run our code given the same data format and computing environment setting. We are happy to answer any questions related to the code for future users who may encounter any issues, as we have previously done with other GitHub repositories.

Reviewers' Comments:

Reviewer #1:

Remarks to the Author:

The manuscript by Cao et al., presents an interesting approach to select 5'UTR sequences to enhance protein production for non-viral gene therapies. I still maintain that the results of the study do not fully support authors claims and the method does not provide enough evidence that it's the engineered 5'UTRs that indeed increase expression of the reporter proteins. The main issue with the method is that there is no control over what and how the mRNA is transcribed or translated from the natural or engineered constructs. Some of examples are:

- It is not clear how authors could argue that all constructs start from the same transcription start site and that they create indeed the predicted 5'UTRs. This is not possible without sequencing mRNAs that are being transcribed from plasmids or viral vectors.
- Do natural or engineered sequences influence in any way strength of transcription from CMV promoter? Do they work as transcription enhancers or superb 5'UTRs? Are these effects associated only with CMV promoter or they work equally well with other promoters?
- How the authors controlled for the translation start sites that are near-cognate GUG, UUG, CUG or other codons. Would authors notice if the protein would have different N-terminals in this case and whether this would impact protein stability and abundance?

These are mostly the issues that are resulting from the approach that uses FACS and DNA sequencing for arguing increase in protein expression associated with mRNA expression and translation.

It is further not clear what would be the effect of codon usage on 5'UTR (line 122) and how is this represented in Supplementary Fig 1?

What is the reason for comparing synthetic 5'UTRs with the vectors with 5'UTRs containing introns in Supplementary Fig. 4? Are authors argue better nuclear export or processing of their synthetic constructs vs intron-containing 5'UTRs?

Finally, authors do not provide reasoning for either the cell choice or selected natural or synthetic constructs tested in the Fig. 3 and 4 (or Supplementary Fig 3). As a matter of fact there are multiple sequences that have higher log₂ fold change (Supplementary Table 4-6) that are not tested (ie. 9937, 4387, 3484, 11227, among the others).

Reviewer #2:

Remarks to the Author:

I thank the authors for providing a revised version of the manuscript as well as a rebuttal that addresses the Reviewer comments. I believe that this revision has strengthened the paper and that overall, this work is technically sound and will interest a range of researchers. However, I believe some additional minor work is needed in order for this manuscript to satisfy its primary goal.

In their rebuttal, the authors have addressed the major concerns. One primary concern was that this work does not provide any mechanistic insight into why specific 5' UTRs perform better than others. The authors have clarified, and I concur, that mechanistic insight is not the goal of this work, but instead that the goal is to provide a novel framework to design, screen, and optimize libraries to enhance expression levels using non-viral gene therapies. The authors have clarified this as their primary goal in the revised manuscript. A second concern was that there is an unsatisfactory amount of detail around the machine learning methods used and the benchmarking that was performed. On this point, there are remaining things that should be addressed, particularly now since the work is being presented as a novel framework for designing, screening, and optimizing libraries to enhance expression levels using non-viral gene therapies that could be

used generally by the community:

1. When building the model to predict translation efficiency (TE) from RNA-Seq and Ribo-Seq data, what considerations need to be made to build an accurate model? The authors explained that they used the default parameters in the randomforest R package, however should this strategy be employed for others who want to design libraries? Specifically here are a few considerations:

- a. Did the authors check for model over-fitting and under-fitting?
- b. Did the authors test a range of number of trees?
- c. Did the authors test a range for the other parameters?
- d. For someone looking to replicate this work with new data, what should their goal be in generating their model?

2. For model evaluation, the authors use the Spearman correlation between the predicted TE and the actual TE. Since this is a ranking metric, it does not necessarily describe the accuracy of the model in predicting TE. To complement this metric, the authors should also perform some sort of error summation. For example, the authors could take different models (i.e. different random forests, linear regression, etc.) and calculate the mean squared error. This would then provide a clear framework for other researchers to follow when building models with other data sets, and have a benchmark for what level of prediction accuracy is necessary for downstream success.

3. Can the authors re-structure their code so that it can be better packaged into a streamlined workflow that could be potentially used by other researchers? If not a generalizable package, a vignette or a more built-out README.md on the github could be a compromise.

4. I thank the authors for providing the github repository. To make this framework usable by the community, I would ask that the authors also include software versions and/or an environment file for code reproducibility.

ANSWERS TO REVIEWERS

Reviewer #1 (Remarks to the Author):

The manuscript by Cao et al., presents an interesting approach to select 5'UTR sequences to enhance protein production for non-viral gene therapies. I still maintain that the results of the study do not fully support the author's claims and the method does not provide enough evidence that it's the engineered 5'UTRs that indeed increase expression of the reporter proteins. The main issue with the method is that there is no control over what and how the mRNA is transcribed or translated from the natural or engineered constructs.

We thank the reviewer for his/her time in going again over our manuscript, and hope that he/she will find that our responses to his individual points raised below will clarify his/her remaining concerns.

The reviewer is concerned that we don't have control over transcriptomic output. However, we should note that even if we would measure transcriptomic levels, the mRNA levels do not necessarily reflect transcriptional strength, because the mRNA levels are also related to the mRNA stabilities. Thus, we would not be able to draw conclusions even if we were measuring mRNA levels.

The main focus of our work is to discover natural or synthetic 5' UTRs that can enhance protein expression regardless of the mechanisms. We believe it is more efficient to do screening and engineering based on the protein expression instead of mechanism-based rational design, because 5' UTRs have complex and multi-level effects on the RNAs. While the reviewer is correct when he/she says that we cannot outrule that changes in 5'UTR primary sequences may affect transcription and/or RNA stability, the final outcome of which 5'UTR sequences lead to enhanced protein expression levels would still be the same, which is the goal of our study. On the other hand, we'd like to note that it is hard to decouple the three processes reliably (transcription, translation and mRNA stability), and the knowledge cannot be directly used to enhance protein expression (which is in fact our goal, not the mechanism). We agree with the reviewer that it would be wonderful to understand with better detail the mechanisms and distinct contributions of how 5' UTR diversity affects protein expression, as this valuable information that can significantly improve our knowledge in RNA biology and eventually benefit RNA rational design. However, these goals are beyond those from our current study. We expect future works will apply our platform as well as other advanced technologies to elucidate how 5'UTR changes affect the transcription, translation and mRNA stability in the future.

Some of examples are:

1 - It is not clear how authors could argue that all constructs start from the same transcription start site and that they indeed create the predicted 5'UTRs. This is not possible without sequencing mRNAs that are being transcribed from plasmids or viral vectors.

We thank the reviewer for his/her question and apologize if this was not sufficiently clear before. The vector used in this study is a commercial plasmid (pVAX1), and the transcription initiation site of the CMV promoter is located in the pVAX1 backbone as shown below, which is the same as for other commercial vectors that contain CMV promoters, such as pcDNA3.1.

We inserted the engineered 5' UTR library upstream of the KOZAK sequence, while the rest leader sequence immediately after CMV promoter was kept the same. The 100 bp 5' UTR candidates diversity regions were inserted before the KOZAK sequence and much after the transcription start site. Thus, all constructs maintain the transcription initiation site untouched.

We have now extended the *Methods* section to include this information, including a new figure (**Supplementary Figure 13**) that illustrates this information (as shown also below for the reviewer's convenience), to ensure that the design of the library is clear to all the readers.

Legend:

Human CMV promoter.

Leader sequence (the full-length 5' UTR)

Transcription start site

5' UTR candidates

Kozak sequence
GFP gene

```
GACATTGATTATTGACTAGTTATTAATAGTAATCAATTACGGGGTCATTAGTTCATAGCCCATATAT
GGAGTTCCGCGTTACATAACTTACGGTAAATGGCCCCGCTGGCTGACCGCCCAACGACCCCCG
CCCATTGACGTCAATAATGACGTATGTTCCCATAGTAACGCCAATAGGGACTTTCCATTGACGTC
AATGGGTGGACTATTTACGGTAAACTGCCCACTTGGCAGTACATCAAGTGTATCATATGCCAAGT
ACGCCCCCTATTGACGTCAATGACGGTAAATGGCCCCGCTGGCATTATGCCCAGTACATGACCT
TATGGGACTTTTCTACTTGGCAGTACATCTACGTATTAGTCATCGCTATTACCATGGTGTATGCGG
TTTTGGCAGTACATCAATGGGCGTGGATAGCGGTTTGACTCACGGGGATTTCCAAAGTCTCCACC
CCATTGACGTCAATGGGAGTTTGTGGCACCAAAATCAACGGGACTTTCCAAAATGTCGTAAC
AACTCCGCCCCATTGACGCAAATGGGCGTAGGCGTGTACGGTGGGAGGTCTATATAAGCAGA
GCTCTCTGGCTAACAGAGAACCCTACTGCTTACTGGCTTATCGAAATTAATACGACTCACTATAG
GGAGACCCAAGCTGGCTAGCGTTAAACTTAAGCTTGGTACCG---5'UTRcandidates---
GCCACCATGGTGAGCAAGGGCGAGGAGCTGTTACCCGGGTGGTGCCCATCCTGGTTCGAGCT
GGACGGCGACGTAACGGCCACAAGTTCAGCGTGTCCGGCGAGGGCGAGGGCGATGCCACCT
ACGGCAAGCTGACCCTGAAGTTCATCTGCACCACCGGCAAGCTGCCCGTGCCCTGGCCACCC
TCGTGACCACCCTGACCTACGGCGTGCAGTGCTTCAGCCGCTACCCCGACCACATGAAGCAGC
ACGACTTCTTCAAGTCCGCCATGCCCGAAGGCTACGTCCAGGAGCGCACCATCTTCTTCAAGGA
CGACGGCAACTACAAGACCCGCGCCGAGGTGAAGTTCGAGGGCGACACCCTGGTGAACCGCAT
CGAGCTGAAGGGCATCGACTTCAAGGAGGACGGCAACATCCTGGGGCACAAGCTGGAGTACAA
CTACAACAGCCACAACGTCTATATCATGGCCGACAAGCAGAAGAACGGCATCAAGGTGAACTTC
AAGATCCGCCACAACATCGAGGACGGCAGCGTGCAGCTCGCCGACCCTACCAGCAGAACACC
CCCATCGGCGACGGCCCGTGTCTGCTGCCCGACAACCACTACCTGAGCACCAGTCCGCCCTG
AGCAAAGACCCCAACGAGAAGCGCGATCACATGGTCTGCTGGAGTTCGTGACCGCCGCCGGG
ATCACTCTCGGCATGGACGAGCTGTACAAGTAA
```

2 - Do natural or engineered sequences influence in any way the strength of transcription from CMV promoter? Do they work as transcription enhancers or superb 5'UTRs? Are these effects associated only with CMV promoters or they work equally well with other promoters?

Differences in 5' UTRs primary sequences and secondary structures may influence protein expression levels via multiple routes and their combinations (transcriptional, translational and mRNA stability) (Refs. 32, 63, PMID: 23832786). For example, in PMID: 23832786 (Dvir, S. et al 2013), the authors specifically mention that: "Accumulating evidence has revealed the importance of 5'-UTRs in shaping eukaryotic protein expression (5–12). Such 5'-UTR sequences encode a variety of cis-regulatory elements, including a 5'-cap structure (13), a translation initiation motif (14–16), upstream AUGs (uAUGs) and upstream ORFs (17, 18), internal ribosome entry sites (19), terminal oligo-pyrimidine tracts (20), secondary structures (21), and G-quadruplexes (22). Many of these elements were shown to control protein levels by altering the efficiency of translation (7–9, 11), whereas some elements affect translation and in addition, transcription (15) or mRNA degradation (23). Despite much progress, significant challenges remain in deciphering the rules by which multiple elements combine to fine-tune protein expression, as well as in quantifying the degree of expression variation that may be jointly explained by known and novel regulatory elements." In another previous work, in Ref. 24 (Weinberger, A. et al. 2013), the authors specifically mention that: "There is no significant correlation between the expression of GFP at mRNA and protein levels, indicating that the increase of protein abundance under the engineered 5'UTRs does not result from mRNA increase." Thus, while it is true that transcription can be affected by 5'UTR sequences, this is typically not what is causing the changes in proteomic outputs. Moreover, the variations of our 5'UTR library have their variation region far from the transcription initiation site and/or promoter (see Diagram in previous comment), and thus, we don't expect transcription to be the major route leading to increased protein outcome.

With regards to the second question of the reviewer, i.e. if we tested other promoters apart from CMV, the short answer is that we did not. The reason why we did not test with promoters other than CMV is because CMV is the most commonly used promoter in plasmids used for gene therapy for high level therapeutic protein expression (e.g. pVAX1, pCMV, pcDNA3.1). In this study, we focused on improving the protein expression levels by expanding the 5' UTR toolbox in mammalian cells, and therefore, we tested the artificial 5'UTRs with CMV promoters, which are the ones used for gene therapy purposes. Our contribution to this system is to expand the 5' UTR toolbox with artificial and shorter 5' UTRs.

However, we did not explore distinct CMV promoters as part of the diversity point of our libraries, since testing the effects of promoter diversity in protein expression levels was not the goal of our study.

3 - How the authors controlled for the translation start sites that are near-cognate GUG, UUG, CUG or other codons. Would authors notice if the protein would have different N-terminals in this case and whether this would impact protein stability and abundance? These are mostly the issues that are resulting from the approach that uses FACS and DNA sequencing for arguing increase in protein expression associated with mRNA expression and translation.

The 5' UTR library diversity region is located before the Kozak sequence. Therefore, the start codon that will be used is the one immediately after the Kozak sequence. Thus, we do not expect that the GFP proteins would have different translation start sites (N-terminal extensions) caused by the diversity introduced in the region that is upstream to the Kozak sequence.

4- It is further not clear what would be the effect of codon usage on 5'UTR (line 122) and how is this represented in Supplementary Fig 1?

We apologize for the confusion. We are not varying the codon usage of the GFP sequence as we are not varying the coding sequence in any region, however we are using the codon usage of the first 15bp of the endogenous sequences as features in our models to predict translation efficiency. To avoid confusion, we have removed "codon usage" from the text, as it is not a "5'UTR feature" per se. Lines 121-123 in the main text now reads as follows:

"Specifically, we extracted sequence features of 5' UTR regions that could be associated with gene expression levels and TE, which included k-mer frequency, RNA folding energy, 5' UTR length, and number of ORFs (Supplementary Fig. 1).

With regards to Supplementary Figure 1, this Figure illustrates the 50 top-ranked features that contributed to the model. Codon usage of the 15 bp in the 5' region of the sequence was not among the top-ranked features, and thus is not shown in the Figure.

5- What is the reason for comparing synthetic 5'UTRs with the vectors with 5'UTRs containing introns in Supplementary Fig. 4? Are the authors arguing for better nuclear export or processing of their synthetic constructs vs intron-containing 5'UTRs?

The main focus of this study is to develop synthetic 5' UTR sequences that can enhance protein expression in mammalian systems, which is a key challenge in biomanufacturing pharmaceutical industry. As there is no approach to predict and rationally design gene constructs to enhance protein drug manufacturing, the most efficient and commonly used approach is to make combinations based on known and good-performing elements, such as CMV and EF1alpha promoters, beta globin intron, intron A and WRPE.

As mentioned in Reference 57 (Xia, W. *et al.*, 2006): "Typical transient gene expression vectors that are currently used only include a strong promoter and an efficient polyadenylation signal to express the gene of interest. Based on the fact that the events downstream of gene transcription are also critical for transient gene expression, inclusion of the post-transcriptional elements (acting on steps downstream of transcription) into expression vectors may largely enhance transient gene expression levels."

Since we aimed to engineer better 5' UTR sequences to enhance protein expression, we compared our synthetic 5' UTRs with the commonly used 5' UTRs, which includes 1 5' UTR element from Ref 56 (Mariati *et al.*, 2010) and 3 introns from Refs 53-55 (Kim, T. J. *et al.*, 2014; Lonza manual; Kang, M. *et al.* 2005). Our results show that the synthetic 5' UTRs designed by us outperformed these introns and TM sequences in protein expression, which means that our strategy is able to discover and engineer novel 5' UTR sequences that can be added to the toolbox of elements with the commonly used introns for biologics production, as they outperform currently used vectors and strategies for gene therapy.

The mechanisms of the intron effects on protein expression are not clear. Better nuclear export or processing might be one important factor in enhancing protein expression, as the reviewer mentions. The reason to include introns was to explore additional diversity to enhance protein expression.

6- Finally, authors do not provide reasoning for either the cell choice or selected natural or synthetic constructs tested in Fig. 3 and 4 (or Supplementary Fig 3). As a matter of fact there are multiple sequences that have higher log₂ fold change (Supplementary Table 4-6) that are not tested (ie. 9937, 4387, 3484, 11227, among the others).

We thank the reviewer for these questions.

With regards to the cell line choice, we chose cell lines that represent several major types used for gene therapy purposes, including: (i) cell lines used in common gene replacement therapies (HEK 293T), (ii) cell lines used for producing DNA vaccines and neutralizing antibodies (RD and C2C12) and (iii) cell lines used for cancer gene therapies (MCF-7). We have now included this information in the manuscript, In Lines 239-243, which now reads, "Finally, we tested how the artificial 5' UTR elements modulate gene expression in different cell types. In addition to HEK 293T, we chose human and mouse muscle cell lines RD and C2C12, respectively, because muscle is a common targeted tissue for vaccines and neutralizing antibody gene therapies^{60,61}. We also selected human breast cancer cell line MCF-7 to test its potential applicability in cancerous cell lines for cancer gene therapies⁶²."

With regards to the choice of selected constructs of 13 to test (In Supplementary Fig. 3, which is Supplementary Fig. 6 now), log₂ fold change was not the only criteria used to select the candidates. Here, we collected cells with high GFP expression into multiple bins. To eliminate the potential false positive hits, we selected the 5'UTRs that have high log₂ fold-changes and p-values of less than 0.05 in ALL three bins. Only 13 out of the 12K library matched the criteria. From the *Methods* section (Lines 140-143): "We chose three sorting brackets of top 0-2.5%, 2.5-5% and 5-10%, instead of only one bracket of 0-10%, to reduce the false positive candidates. Only the candidates that were significantly over-represented in all three bins, relative to background (0-100% bin), were selected as candidates for further validation."

With regards to Figure 4, we chose the top three 5' UTR constructs out of the 13 tested above for subsequent studies and optimization. This is mentioned in Lines 202-205 in the manuscript text: "From the six experimentally validated 5' UTRs that increased GFP production in HEK 293T cells (Fig. 4B), we chose the top three 5' UTRs candidates for additional testing in human muscle cells, since DNA-encoded therapeutics are often delivered into human muscle to trigger the expression of vaccines or therapeutics (Fig. 4C)."

Reviewer #2 (Remarks to the Author):

I thank the authors for providing a revised version of the manuscript as well as a rebuttal that addresses the Reviewer comments. I believe that this revision has strengthened the paper and that overall, this work is technically sound and will interest a range of researchers. However, I believe some additional minor work is needed in order for this manuscript to satisfy its primary goal.

We thank the reviewer for his/her time in going again over our manuscript, and hope that he/she will find that the additional work we have put in improving/packaging our code will now make it more user-friendly by the community, which we agree is of major importance.

In their rebuttal, the authors have addressed the major concerns. One primary concern was that this work does not provide any mechanistic insight into why specific 5' UTRs perform better than others. The authors have clarified, and I concur, that mechanistic insight is not the goal of this work, but instead that the goal is to provide a novel framework to design, screen, and optimize libraries to enhance expression levels using non-viral gene therapies. The authors have clarified this as their primary goal in the revised manuscript. A second concern was that there is an unsatisfactory amount of detail around the machine learning methods used and the benchmarking that was performed. On this point, there are remaining things that should be addressed, particularly now since the work is being presented as a novel framework for designing, screening, and optimizing libraries to enhance expression levels using non-viral gene therapies that could be used generally by the community:

1. When building the model to predict translation efficiency (TE) from RNA-Seq and Ribo-Seq data, what considerations need to be made to build an accurate model? The authors explained that they used

the default parameters in the randomforest R package, however should this strategy be employed for others who want to design libraries? Specifically here are a few considerations:

a. Did the authors check for model over-fitting and under-fitting?

We performed model cross-validation, specifically using 10-fold cross-validation. This information is included now in the GitHub repository. Based on Supplementary.Figs.3-5, our random forest performs much better than other methods in cross-validation setting, so it is less likely under-fitting or over-fitting.

Supplementary Figure 3. The prediction comparisons between different models in HEK 293T cells. A) TE prediction spearman correlation; B) TE prediction R²; C) RNA RPKM prediction spearman correlation; C) RNA RPKM prediction R².

Supplementary Figure 4. The prediction comparisons between different models in PC3 cells. A) TE prediction spearman correlation; B) TE prediction R²; C) RNA RPKM prediction spearman correlation; D) RNA RPKM prediction R².

Supplementary Figure 5. The prediction comparisons between different models in muscle cells. A) TE prediction spearman correlation; B) TE prediction R²; C) RNA RPKM prediction spearman correlation; D) RNA RPKM prediction R².

b. Did the authors test a range of number of trees?

We did not systematically check for a range of parameters or number of trees; however, these parameter tunings and additional options could be incorporated by the future users, now that we have made the code more streamlined and packaged following the reviewer's suggestion.

c. Did the authors test a range for the other parameters?

We initially had tested other algorithms apart from Random Forest (RF), however, we found they performed worse than RF. We have now included the option of additional algorithm testing as part of the pipeline (option `--model`) such that the user can choose to run `glmnet`, `Rpart` and `SVM`, in addition to Random Forest.

d. For someone looking to replicate this work with new data, what should their goal be in generating their model?

The goal would be to generate a model (and thus synthetic sequences) that is reflective enough of the features of the endogenous sequences (i.e. not to put too much/too little folding energy in the 5'UTR region, avoid k-mers that may be completely under-represented in biological sequences that could lead to undesired recruitments of proteins, etc) while exploring novel diversity via the genetic algorithm that evolves the sequences. This model will be different for each species, as their 5'utr endogenous sequences have distinct features, k-mer frequencies, etc. Thus, the user can now use it to generate novel sequences that will fit those of the species of their interest.

For example, the work by Kudla et al (Science 2009: <https://pubmed.ncbi.nlm.nih.gov/19359587/>) generated a set of synthetic sequences (in this case they were manipulating the coding sequence) to explore the effect of codon usage in GFP expression levels. However, they ended up generating a set of sequences that were actually causing that the folding energy of the region close to the start codon was completely out of the range of folding energies that endogenous sequences had, making the actual GFP variants unusable in terms of revealing the role of codon usage in determining/enhancing protein expression levels. This was indeed shown later by Tuller *et al.*

(<https://www.pnas.org/content/107/8/3645>) who showed that their GFP variants in the Kudla et al manuscript contained folding energies that were beyond those observed in endogenous sequences.

Thus, the goal from someone using our algorithm would be to generate a set of 5'UTR sequences in their species of interest, that would fit the "general features" of endogenous sequences, while exploring and generating novel sequence diversity.

2. For model evaluation, the authors use the Spearman correlation between the predicted TE and the actual TE. Since this is a ranking metric, it does not necessarily describe the accuracy of the model in predicting TE. To complement this metric, the authors should also perform some sort of error summation. For example, the authors could take different models (i.e. different random forests, linear regression, etc.) and calculate the mean squared error. This would then provide a clear framework for other researchers to follow when building models with other data sets, and have a benchmark for what level of prediction accuracy is necessary for downstream success.

Following the reviewer's suggestion, the pipeline (https://github.com/zzz2010/5UTR_Optimizer/blob/master/README.md) can now build models with 4 different machine learning algorithms (RF, `glmnet`, `Rpart` and `SVM`), as showed in Supplementary Figs 3-5.

In addition, following the reviewer's suggestion, the step #3 (model evaluation) now reports root mean squared error (rsq) in addition to Spearman correlation.

We added Supplementary Figs 3-5, showing the performance difference across random forest, regularized linear regression (based on `glmnet`), `SVM`, regression tree in the 10-fold cross validation setting.

3. Can the authors re-structure their code so that it can be better packaged into a streamlined workflow that could be potentially used by other researchers? If not a generalizable package, a vignette or a more built-out README.md on the github could be a compromise.

Following the reviewer's suggestion, we have now packaged and streamlined the workflow. Details can be found in the GitHub README file (https://github.com/zzz2010/5UTR_Optimizer/blob/master/README.md).

Briefly, we have created a master program, run_pipeline.py, that now performs all 4 different steps:

- Step 1. Feature extraction from endogenous sequences → run_pipeline feature_extract
- Step 2. Model generation using extracted features to predict Translation Efficiency (TE) → run_pipeline model_build
- Step 3. Model evaluation → run_pipeline model_eval
- Step 4. Generation of novel 'evolved' sequences using a Genetic Algorithm → run_pipeline sequence_generate

4. I thank the authors for providing the github repository. To make this framework usable by the community, I would ask that the authors also include software versions and/or an environment file for code reproducibility.

Following the reviewer's suggestion, we have added a section in the GitHub repository listing all the software versions used (https://github.com/zzz2010/5UTR_Optimizer/blob/master/README.md#Dependencies-and-versions). In addition, we have also added an environment file to the repository (environment.yml) that can be used to recreate a conda environment with all the dependencies (conda env create -f environment.yml).

Reviewers' Comments:

Reviewer #1:

Remarks to the Author:

The revised manuscript by Cao et al., did improve on the clarity but I still have concerns about the missing controls in their approach and experiments that are based only on FACS sorting and sequencing of DNA (and not RNA).

Authors main claim is that they are interested only in the outcome of which engineered sequence enhances protein expression levels in this study (regardless of the mechanism). However, the main point of my previous remarks is that if engineered sequences in their system work as downstream transcriptional enhancers by simply increasing transcription then sequences are not fulfilling their role as enhanced 5'UTRs (title of the paper) but transcriptional enhancers. This is the premise of the study and analysis of mRNA levels and translational efficiency (TE) written in introduction and first section of results. Simple experiment, like use of different promoter or use of in vitro transcribed mRNAs (shown for example in Fig 4C-F or 5B) followed by the measurement of reporter expression or qRT-PCR analyses would indicate that this is not the case and that effects seen in the Fig 4 and 5 are not promoter specific. This would improve manuscript and would substantiate authors claim about enhanced 5'UTRs.

More generally, applying protocols for either 5'-CAGE (Takahashi et al., 2012), RNA-Seq and Ribo-Seq (which are used as trained data) on FACS sorted cells would give even better insight. However I understand that these protocols would require additional time and expertise. Such experiments would give more mechanistical insight as well which authors argue is not a goal of this study.

Reviewer #2:

Remarks to the Author:

I thank the authors for providing a revised version of the manuscript as well as a rebuttal that addresses the comments and questions poised by the reviewers. I believe that this revision has continued to strengthen the paper and that overall, this work is technically sound and will interest a range of researchers.

The primary goal of this work is to provide a novel framework to design, screen, and optimize libraries to enhance expression levels using non-viral gene therapies. Although there is additional work that could be done outside of this goal (reflected in our initial review and in the comments of reviewer #1), I believe that the authors have clearly and logically addressed the comments and concerns that are in relation to the primary goal of this manuscript, and it is therefore suitable for publication.

ANSWERS TO REVIEWERS

Reviewer #1 (Remarks to the Author):

The revised manuscript by Cao et al., did improve on the clarity but I still have concerns about the missing controls in their approach and experiments that are based only on FACS sorting and sequencing of DNA (and not RNA).

We thank the reviewer for his/her time in going again over our manuscript and hope that he/she will find that our responses to his/her individual points raised below will clarify his/her remaining concerns.

Authors main claim is that they are interested only in the outcome of which engineered sequence enhances protein expression levels in this study (regardless of the mechanism). However, the main point of my previous remarks is that if engineered sequences in their system work as downstream transcriptional enhancers by simply increasing transcription then sequences are not fulfilling their role as enhanced 5'UTRs (title of the paper) but transcriptional enhancers. This is the premise of the study and analysis of mRNA levels and translational efficiency (TE) written in introduction and first section of results. Simple experiment, like use of different promoter or use of in vitro transcribed mRNAs (shown for example in Fig 4C-F or 5B) followed by the measurement of reporter expression or qRT-PCR analyses would indicate that this is not the case and that effects seen in the Fig 4 and 5 are not promoter specific. This would improve manuscript and would substantiate authors claim about enhanced 5'UTRs.

We thank the reviewer for the comment. Our study was inspired by the remaining technical challenges of the low drug production in gene therapies. Genetic elements, such as 5' UTRs, can enhance protein expression and reduce the doses of the genetic loads (DNA or RNA drugs) as well as the amounts of toxic delivery materials (e.g., lipid nanoparticles). The goal of this study is to discover and engineer improved 5'UTRs elements for them. 5' UTRs can enhance protein expression by a variety of ways, such as increasing transcriptional efficiency (TE), translational efficiency and mRNA stability. TE is one of the key factors that influence protein expression. We agree with the reviewer that testing with different promoters would provide better insight on the mechanisms underlying the 5' UTRs, however, the goal of this study was not to reveal the mechanism, but to obtain novel plasmids that enhance protein expression levels.

More generally, applying protocols for either 5'-CAGE (Takahashi et al., 2012), RNA-Seq and Ribo-Seq (which are used as trained data) on FACS sorted cells would give even better insight. However, I understand that these protocols would require additional time and expertise. Such experiments would give more mechanistical insight as well which authors argue is not a goal of this study.

We agree with the reviewer that the suggested experiments would provide better insights on understanding the mechanisms of the 5' UTR candidates found in this study and how 5' UTRs influence on gene expression in general. We believe mechanistical insight will guide versatile 5' UTR optimization beyond enhancing protein expression in future works.

Reviewer #2 (Remarks to the Author):

I thank the authors for providing a revised version of the manuscript as well as a rebuttal that addresses the comments and questions posed by the reviewers. I believe that this revision has continued to strengthen the paper and that overall, this work is technically sound and will interest a range of researchers.

The primary goal of this work is to provide a novel framework to design, screen, and optimize libraries to enhance expression levels using non-viral gene therapies. Although there is additional work that could be done outside of this goal (reflected in our initial review and in the comments of reviewer #1), I believe that the authors have clearly and logically addressed the comments and concerns that are in relation to the primary goal of this manuscript, and it is therefore suitable for publication.

We thank the reviewer for his/her time and comments, and for helping us improve the overall quality of the manuscript throughout the review process with his/her suggestions.